# Classification of the Crosslink Density Level of Para Rubber Thick Film of Medical Glove by Using Near-Infrared Spectral Data

**DOI:** 10.3390/polym16020184

**Published:** 2024-01-08

**Authors:** Jiraporn Sripinyowanich Jongyingcharoen, Suppakit Howimanporn, Agustami Sitorus, Thitima Phanomsophon, Jetsada Posom, Thanapol Salubsi, Adisak Kongwaree, Chin Hock Lim, Kittisak Phetpan, Panmanas Sirisomboon, Satoru Tsuchikawa

**Affiliations:** 1Department of Agricultural Engineering, School of Engineering, King Mongkut’s Institute of Technology Ladkrabang, Bangkok 10520, Thailand; jiraporn.jo@kmitl.ac.th (J.S.J.); 64601240@kmitl.ac.th (S.H.); thitimap.june@gmail.com (T.P.); panmanas.si@kmitl.ac.th (P.S.); 2National Research and Innovation Agency (BRIN), Jakarta Pusat 10340, Indonesia; 3Department of Agricultural Engineering, Faculty of Engineering, Khon Kaen University, Khon Kaen 40002, Thailand; 4W. A. Rubber Mate Co., Ltd., Bangkok 10240, Thailand; sorah.salabsee@gmail.com (T.S.); a.kongwaree@gmail.com (A.K.); 5Thai Rubber Latex Group Public Co., Ltd., Chonburi 20190, Thailand; lim@thaitex.com; 6Department of Engineering, King Mongkut’s Institute of Technology Ladkrabang, Prince of Chumphon Campus, Chumphon 86160, Thailand; kittisak.ph@kmitl.ac.th; 7Graduate School of Bioagricultural Sciences, Nagoya University, Nagoya 464-8601, Japan; st3842@agr.nagoya-u.ac.jp

**Keywords:** crosslink density, natural rubber, medical glove, near infrared, linear tunable filter, MicroNIR

## Abstract

Classification of the crosslink density level of para rubber medical gloves by using near-infrared spectral data combined with machine learning is the first time reported in this paper. The spectra of medical glove samples with different crosslink densities acquired by an ultra-compact portable MicroNIR spectrometer were correlated with their crosslink density levels, which were referencely evaluated by the toluene swell index (TSI). The machine learning protocols used to classify the 3 groups of TSI were specified as less than 80% TSI, 80–88% TSI, and more than 88% TSI. The 80–88% TSI group was the group in which the compounded latex was suitable for medical glove production, which made the glove specification comply with the requirements of customers as indicated by the tensile test. The results show that when comparing the algorithms used for modeling, the linear discriminant analysis (LDA) developed by 2nd derivative spectra with 15 k-best selected wavelengths fairly accurately predicted the class but was most reliable among other algorithms, i.e., artificial neural networks (ANN), support vector machines (SVM), and k-nearest neighbors (kNN), due to higher prediction accuracy, precision, recall, and F1-score of the same value of 0.76 and no overfitting or underfitting prediction. This developed model can be implemented in the glove factory for screening purposes in the production line. However, deep learning modeling should be explored with a larger sample number required for better model performance.

## 1. Introduction

The US Food and Drug Administration [1] reported: Due to the problem of the COVID-19 epidemic, there is a high medical demand, so a large amount of public health equipment has to be produced, especially medical rubber gloves that must be used to prevent direct contact with patients. And it is also used only once and then discarded. In the production process, there must be a process that must be controlled. Usually, medical gloves are divided into surgical gloves, which are long from the hand to the elbow and are rubber gloves that must be sterilized, light-weight, and highly flexible. The other type is used for general examinations (medical examination gloves), which are about the length of the wrist, highly flexible, and used to prevent contact in general disease examinations. Both types are single-use medical examination gloves. Gloves shall be manufactured from compounded natural rubber, nitrile rubber, or polychloroprene rubber latex in accordance with International Standard ISO 11193-1 [2].

The production of rubber gloves requires the preparation of a latex compound for use in dipping molding. The latex compound must be vaporized with chemicals and additives to meet the standards set for specific types of gloves. When chemicals are added to the latex, they will interact, causing the separated rubber molecules to coagulate. Together, they form a polymer chain, forming a crosslink. In production, the amount of this crosslink affects the properties of rubber gloves. The manufacturing company must configure the appropriate crosslink density for quality production.

Criteria for classification of the crosslink density levels depend on the manufacturer and product standards. The examination gloves of the W.A. RUBBER MATE CO., LTD., Thailand, who provided the chemical for natural rubber latex compounding for the production of gloves used in this research, indicated the usable range is 80–88% toluene swell in-dex, which is an appropriate crosslink density value for the compound latex for the production of medical rubber gloves.

Toluene swell index (TSI) or equilibrium swelling values are determined by immersing a latex film in the solvent, usually toluene, and measuring the increase in linear dimensions at equilibrium [3]. TSI is directly related to the crosslink density, but the procedure is relatively slow and requires toluene, a hazardous chemical, and the total time for the test, including drying of the film, is about 1 h or more [3].

Near-infrared spectroscopy is a non-destructive technique that requires no chemicals, making it environmentally friendly. It is a rapid method that requires only 1–2 min to scan the sample and calculate the interested parameter or constituent. The accuracy of the NIR technique prediction is comparable to the reference test, but not more.

Since NIR spectra do not carry a signal to detect C–S bonds, which are the cross-link between isoprene units [4], the absorption of C–H bonds, which were changed during crosslinking, made the NIR spectroscopy model workable, including the classification of crosslink density levels in compound latex.

Machine learning algorithms such as linear discriminant analysis (LDA), support vector machine (SVM), artificial neural network (ANN) and k-nearest neighbor (kNN) combined with NIR spectral data have been used in classification of rubber objects e.g., by using NIR spectra 900–2500 nm, the k-NN classifiers (1-NN with standard normal variate (SNV) data pre-treatment and variable selection based on F-test-statistics (mean class-specific accuracy (MCSA) = 0.78), SVM with SNV data pre-treatment and PCA di-mension reduction (mean MCSA = 0.85), and LDA with SNV without any variable selec-tion or dimension reduction (MCSA = 0.86) in classification of historical and modern polymers including natural rubber where 20% of erroneous predictions were due to its misidentification as a man-made rubber polyisoprene which has a similar composition [5]. There was none of research reports on rubber gloves quality classification and the use of low cost portable NIR spectrometers including ultra compact MicroNIR spectrometer (Viavi, Chandler, AZ, USA) which has been used for prediction of bamboo biomass activation energy (Ea) by PLS modeling for Ea at n = 1 and Ea at n ≠ 1 showed coefficients of determination of 0.781 and 0.714, respectively [6], identifying illicit drugs in oral fluids, including cocaine (COC), amphetamine (AMP), and Δ9-tetrahydrocannabinol (THC) where the partial least squares–discriminant analysis (PLS-DA) prediction for COC, AMP, and THC provided the non-error rate (NER, %) of prediction for all the processed classes (NER not less than 90%) resulting in 100% correctly classified samples when predicting illicit-drug abuse [7] and identification of marine macro-, meso-, and microplastic litter collected on beaches in sediments and seawater and enabled the correct identification of marine plastic litter for macro-, meso- (96%), and microplastics (73%) with exception of totally black items and items less than 1 mm in size [8] and more. Therefore, the objectives of this research were to report for the first time the feasibility of using NIR spectroscopy in the classification of crosslink density levels of medical gloves when using the NIR spectra of thick films of medical gloves scanned by an ultra-compact diffuse reflectance MicroNIR spectrometer (960–1650 nm) combined with machine learning.

## 2. Materials and Methods

### 2.1. Sample Preparation

Concentrated natural rubber latex used for compounding for molding medical gloves was obtained from the factory of Thai Rubber Latex Group Public Company Limited in Nongyai, Chonburi, Thailand, and the chemicals used in producing the thick film of medical gloves, including Potassium Hydroxide (KOH), Hydroperse (WA-4C), Octolite (WL-M), Hydrocal 295 (Filler), coagulant used, and mixing ingredients formula for manufacturing medical gloves, were provided by W.A. Rubber Mate Co., Ltd. in Nongmaidaeng, Chon Buri, Thailand, specifically. The concentrated latex was mixed with chemicals and stirred for up to 90 h for curing. The sampling for measuring crosslink density was first started at 24 h and then every 3 h for 90 h; 31 samples were obtained. The experiment was carried out in five rounds. Therefore, 155 samples were obtained. For the measurement of the crosslink density of the thick film of medical gloves, an aluminum sheet of 90 × 220 × 0.5 mm was used as a mold. The mold was dipped in coagulant and placed in the oven (MEMMERT UF 30, Schwabach, Germany) at 100 °C for 10 min. Then, the mold was dipped in the cured sample at the predetermined curing time (from 24 h to 90 h) described above for 10 s and dried in the oven at 100 °C for 30 min to obtain the rubber-thick film of the medical glove for further experiments. Then, the thick rubber film of the medical glove was removed from the aluminum sheet by hand, and during the removal, corn starch was applied on both sides to prevent the rubber film from sticking together.

The total real number of spectra after outlier elimination in this study is 130, of which 93 (42:29:22) were used as training and 37 as testing. With the Synthetic Minority Over-sampling Technique (SMOTE), the total spectra were increased to 163, of which 126 (42:42:42) were used as training and 37 real spectra were used as testing. The sample collection duration of our experiment was during 28 January 2022 till 9 January 2023 which is the year-round production of the factory, confirming the robustness of the calibration model’s wide variation data.

### 2.2. NIR Spectroscopy

Ultra compact NIR spectrometer with a wavelength range of 900–1700 nm (MicroNIR Pro 1700ES Spectrometer, VIAVI Solutions Inc., San Jose, CA, USA) was used for the absorbance spectrum of the thick film sample acquisition. The scanning resolution was 6.2 nm. Therefore, there were 125 points (908–1676 nm) obtained to form the spectrum. The white reference spectrum and dark reference spectrum for the background compensation were scanned at the beginning of every 10 samples to be scanned. The white reference material was Spectralon^®^ and the dark reference spectrum was obtained by scanning the flour from a height of ~60 cm.

### 2.3. Toluene Swelling

The cross-link density of the compound latex is measured using the Toluene Swell Index (TSI). Cut the thick film of medical glove obtained in 2.1 into a circle with a diameter of 25 mm by die cutting and submerge it in toluene in a glass Petri disk with a 100 mm diameter and 15 mm height with a glass cover. The Petri disk was placed on the squared paper for 10 min. The experiment was carried out in triplicate per sample. By swelling in the toluene solution, the amount of crosslink density can be determined. The swelling value gradually decreased as more cross-linking bonds were formed. The swelling of the rubber-thick film was measured by reading the value before and after 10 min from the scale of the squared paper. The toluene swell index (TSI) was calculated by Equation (1).
TSI (%) = (Y − X)/X × 100(1)

Y is the diameter of the circular film submerged in toluene for 10 min, and X is the diameter of the circular film before being submerged in toluene.

The crosslink density of medical rubber glove production is highest when TSI is in the range of 80–88%. Therefore, the TSI levels in this study were at three levels, including less than 80, 80–88, and more than 88%.

### 2.4. Near Infrared Spectroscopy Classification Modelling

Figure 1 shows the schematic of the developed classification model.

#### 2.4.1. Spectral Pretreatment

After spectrum acquisition, the obtained spectra were subjected to mathematical pretreatment to reduce interference from an unstable environment and sample influence on the spectra, which developed noise, baseline shift, and tilting problems. Before spectral pretreatment, the abnormal spectra observed by the eyes were eliminated, and 42, 29, and 22 samples of the TSI levels of less than 80, 80–88, and more than 88%, respectively, were obtained for the calibration set. The prediction set included 17, 12, and 8 samples of TSI levels of less than 80, 80–88, and more than 88%, respectively. Therefore, 130 is the total number. Then, the principal component analysis was applied to the raw spectra, and the three principal component scores and x-loading were plotted.

The pretreated methods of Savitzky–Golay smoothing, which resulted in reduced noise in the spectrum, were first applied, and then the second derivative (segment size 21) [9], multiplicative scatter correction (MSC), standard normal variate (SNV), detrending, and normalization were applied consecutively [10].

Other preprocessing resulting from normalization, including min-max normalization, is also considered in this study. In addition to that, robust normal variate (RNV) preprocessing is also used in this study to handle light scatter effects like SNV. If the SNV formula is subtracted by the mean, RNV is subtracted by the median of each spectral variable and subsequently divided by the standard deviation of the spectrum. Like SNV, RNV and L2 norm scaling can also handle scatter problems on the spectrum [11]. Finally, log transformation preprocessing is also used to scale and transform NIR spectra to increase the relationship between absorbance and response to become linear again [12].

SMOTE involves oversampling through the creation of synthetic new samples based on the original ones. These new samples are generated using random constants within the range of [0,1] and are placed at distances determined by the original samples [13]. As a result, SMOTE preprocessing typically does not introduce noise unless the original samples contain noise, in which case the new spectra might inherit noise. To address this issue, we employed Savitzky–Golay smoothing, which helped mitigate the problem.

#### 2.4.2. Classification Analysis

With full and selection wavelengths, after performing several spectral pretreatments, the classify models were developed using supervised machine learning classification algorithms, including 2 algorithms as linear classifiers (LDA, KNN) and 2 algorithms as hybrid classifiers (linear and nonlinear), including SVM and ANN. The wavelength was selected using the k-best and genetic algorithm (GA) methods. k-Best selection of wavelengths is a spectral analysis technique that selects the most informative wavelengths from a dataset based on specific criteria, enhancing computational efficiency and reducing noise. k-Best directly identifies the best “k” individual wavelengths based on specific criteria, resulting in a subset of fixed size. In this study, we used the function select k-Best, including the ANOVA F-value between features for classification tasks (“f_classif”), mutual information for a discrete target (“mutu-al_info_classif”), and chi-squared statistics of non-negative features for classification tasks (“chi2”) with a range of top features to select (k) between 1 and 50 [14]. GA was originally proposed by Holland in 1975 and refers to the natural selection and genetic mechanisms in the biological world. According to Chu et al. [15], the realization of GA mainly includes five basic elements: parameter coding, initialization of the population, design of fitness function, genetic operation design, convergence criterion, and selection of variables. For dimensional reduction, this study will use a PCA algorithm with just 10 PCs, which will be considered later as feature variables.

The first time, the existing data set of samples was split into 2 groups, including the calibration set (70% of all samples) and the validation set (30% of all samples). After that, pretreatment, modeling, and testing of the modeling performance will be conducted. The second time, because this study has imbalanced data, the data augmentation technique will be used by applying the SMOTE to a calibration data set. This technique has been reported several times in research papers as being able to improve the performance of the calibration model being generated, especially when using NIR spectroscopy data for classification problems [16,17,18]. A full description of this method can be read in the Brownlee [19] short report. The data ratio on calibration with 3 groups before augmentation is 42:29:22. By conducting the SMOTE method, the calibration data will generate a ratio of 42:42:42. After that, the pretreatment will be performed by modeling and testing the model using the validation data set.

In order to identify an appropriate model for classifying the crosslink density level of para rubber medical gloves, the supervised machine learning classification algorithms, including Artificial Neural Networks (ANN), Support Vector Machines (SVM), k-Nearest Neighbors (kNN), and Linear Discriminant Analysis (LDA), consider the distinct strengths exhibited by each algorithm type.

Artificial Neural Networks (ANN) are a form of deep learning that models the neural structure of the human brain [20]. They consist of interconnected nodes (neurons) organized into layers (input, hidden, and output) [21]. ANN learns by adjusting weights in response to input data, aiming to map inputs to outputs through a training process, commonly employing algorithms such as backpropagation [21]. ANN demonstrates the ability to comprehend complex patterns and adaptability [22].

Support Vector Machines (SVM) are supervised learning algorithms. They seek a hyperplane that effectively segregates data into distinct classes, aiming to maximize the margin (distance) between the closest points of different classes, referred to as support vectors, for accurate classification of new data points [23]. SVM proves effective in high-dimensional spaces [24].

k Nearest Neighbors (kNN) is a straightforward, instance-based learning algorithm utilized for classification. It predicts outcomes based on the majority class or average value of the k nearest data points to a query point in the feature space [25]. Being non-parametric, it requires no training [26].

Linear Discriminant Analysis (LDA) is a technique used for both dimensionality reduction and classification [27]. It aims to determine linear combinations of features that effectively differentiate classes within a dataset [28]. LDA projects data onto a lower-dimensional space, maximizing the distance between class means and minimizing the variance within each class [28]. It functions as a dimensionality reduction method and effectively handles multi-class problems.

The calibration set was used to validate the machine learning estimators, and the op-timum hyperparameter of each ML algorithm was found with the “GridSearchCV” command of the Scikit-learn module. Table 1 presents predefined parameters for performing the “GridSearchCV” command. The optimum hyperparameter was searched based on the highest cross-validation accuracy from executing the 5-fold cross-validation. The process of modeling was implemented in the Python (3.11.4) language with the machine learning packages of Scikit-learn (1.2.2) and the programming tool of Jupyter Notebook (6.5.4).

### 2.5. Classification Model Performance Determination

The confusion matrix represents the resulting predicted classes of the model, used to determine the classification performance [20]. It consists of true positive (TP): number of samples correctly predicted as positive; false negative (FN): number of samples wrongly predicted as negative; false positive (FP): number of samples wrongly predicted as positive; and true negative (TN): number of samples correctly predicted as negative (Figure 2).

The classification model performance was defined by classification accuracy, weighted average of precision, weighted average of recall, and weighted average of F1-score calculated as illustrated in Table 2.

### 2.6. Validation by Unknown Real Sample Set form Factories

Medical rubber gloves without powder were collected from 4 factories (4 gloves per factory). Therefore, 16 gloves. Each glove was scanned by placing a MicroNIR window on the intact glove (a two-layer scan) placed on the aluminum plate as a reflector, making this scanning the transflectance mode. The scanning was conducted with 5 scans per position and 4 positions on one glove. Therefore, there are 320 spectra in total. Then, the MicroNIR was inserted inside the glove and scanned only one layer using the same procedure. After that, the glove was subjected to the TSI test immediately.

Every spectrum scanned was subjected to some models developed, including LDA, kNN, and SVM, with different pretreatment methods, and the model classification performance was calculated.

## 3. Results

### 3.1. NIR Spectra of Medical Glove Samples Measured by MicroNIR Spectrometer

Figure 3a–j shows raw (a), Savitzky–Golay smoothing (b), 2nd derivative (c), multiplicative scatter correction (MSC) (d), standard normal variate (SNV) (e), detrending (f), min-max scaling (g), robust normal variate (RNV) (h), log transform (i), and L2 norm scaling (j) spectra of different levels of toluene swell index of 93 glove samples of 3 levels of TSI, respectively.

Figure 4 and Figure 5 show the 3D score plots and the X-loading curves from PCA of full spectra, respectively. From Figure 4, by PC1, the film sample with a TSI greater than 88% was more or less separated from other groups, while the samples with a TSI less than 80% and a TSI of 80–88% were mixed.

### 3.2. Statistics of Toluene Swell Index of Thick Film Samples for Modelling

Table 3 shows the number of toluene swell index (TSI) of glove samples of different levels for model development before and after SMOTE.

### 3.3. Performance of Classification Models

#### 3.3.1. Full Spectra

The results of classifying three different TSI levels of thick-film medical gloves for the calibration and prediction sets in an imbalanced dataset are presented in Table 4, which shows the LDA achieved the highest accuracy (0.99) during calibration, but when validated, it exhibited the lowest accuracy (0.46). In contrast, other models demonstrated similar performance for both the calibration and validation sets. The calibration accuracies of ANN, SVM, and kNN were 0.70, 0.74, and 0.77, respectively, while the validation set accuracies were 0.84, 0.70, and 0.70, respectively. For LDA, though the accuracy in calibration was the highest, the accuracy in validation was the lowest, indicating the overfit of the model that occurred due to the sample size not significantly conforming to the number of hyperparameters tuned [30,31] and to the effect of the separation method, which caused the distribution of the pretreated spectra of the calibration set and validation set to be different. This can be used as a rationale for the LDA models in Table 5, Table 6 and Table 7.

After generating the data using SMOTE, the results of classifying three different TSI levels of thick-film medical gloves for the calibration and prediction sets with a balanced dataset are presented in Table 5. Although the issue of imbalanced data has been addressed, the accuracy has remained the same. The calibration accuracies for ANN, SVM, kNN, and LDA were 0.78, 0.78, 0.75, and 0.99, respectively, while the validation set accuracies were 0.73, 0.73, 0.73, and 0.43, respectively.

However, the F1-scores of G2 and G3 (groups with smaller sample sizes than G1) were higher after SMOTE (Balanced data).

#### 3.3.2. Selective Spectra by k-Best and GA Method

The results of classifying three different TSI levels of thick-film medical gloves for the calibration and prediction sets in an imbalanced dataset with selected wavelengths are presented in Table 6, where the k-Best and GA methods reduced the number of wavelengths from 125 (full wavelength) to 14–62 wavelengths, but the accuracy of the model remained the same, indicating other bands that were not featured wavelengths had a neutral effect on classification. The calibration accuracies for ANN, SVM, and kNN with k-Best were 0.74, 0.78, and 0.75, respectively, while the validation set accuracies were 0.68, 0.65, and 0.68, respectively. The calibration accuracy for LDA with k-Best (15 wavelengths) decreased to 0.72 (from 0.99), but the validation accuracy increased from 0.43 to 0.76.

The calibration accuracies for ANN, SVM, kNN, and LDA with GA were 0.71, 0.74, 0.76, and 0.91, respectively, while the validation set accuracies were 0.65, 0.70, 0.70, and 0.65, respectively.

Similar to the full spectra models, the results of classifying the three different TSI levels of thick-film medical gloves for the calibration and prediction sets after balancing the dataset by SMOTE with selected wavelengths were similar to the results before balancing the dataset (Table 7). However, the F1-scores of G2 and G3 were higher. The k-Best and GA methods reduced the number of wavelengths from 125 (full wavelength) to 1–74 wavelengths. The calibration accuracies for ANN, SVM, kNN, and LDA with k-Best were 0.70, 0.79, 0.80, and 0.75, respectively, while the validation set accuracies were 0.70, 0.65, 0.68, and 0.70, respectively. The calibration accuracies for ANN, SVM, kNN, and LDA with GA were 0.80, 0.78, 0.75, and 0.99, respectively, while the validation set accuracies were 0.73, 0.70, 0.73, and 0.57, respectively.

#### 3.3.3. Dimensional Reduction by PCA

The results of the classification of three different TSI levels of thick film in medical gloves for the calibration and prediction sets for the imbalance dataset and after balancing the dataset by SMOTE with the independent variable dimension reduction using PCA are presented in Table 8 and Table 9, respectively. The LDA with Savitzky–Golay smoothing + detrending and + Robust normal variate for the PCA data before and after SMOTE, respectively, show the best prediction performance indicated by accuracy, precision, recall, and F1-score of 0.75, 0.75, 0.76, and 0.75 and 0.76, 0.75, 0.76, and 0.75, respectively. From Figure 4, PC1 has covered the highest spectral informative variance of 94.42%, leaving only 4.77 and 0.63% for PC2 and PC3, and 0.18% for PC4 and PC125. However, when the PC scores of different PCs were used for model development, PC1 was the main PC and could only be classified with a performance of 0.75–0.76, indicating the highest spectral informative variance, but there was just a moderate correlation between those spectra.

These performance indicators, together with the X-loading loading of PC1 to PC3 (Figure 5), indicated a moderate relationship between the PC score of NIR spectral data obtained by PCA and TSI. From Figure 5, the high peaks of cis-1,4-polyisoprene from pure Para rubber sheet at 1200, 1390, and 1420 nm, as indicated by Sirisomboon et al. [5], were shown in the X-loading of PC1 to PC3, which confirmed that there was a relationship between the NIR vibration of natural rubber and the related property, which in this case was TSI but moderate.

#### 3.3.4. Validation Result by Unknown Real Sample Set from Factories

Table 10 shows the production information of the unknown glove samples and the TSI value of every sample. The models developed were used to predict the TSI value of the glove sample. The TSI of every glove was 60% except one, which was 72%, which was in group 1 (<80%). This indicated the uniformity of glove production. Though there may be an opportunity to have an out-of-access group indicating the need for non-destructive detection of the product in real-time online for 100% cross-density level detection.

By the TSI test, the results show that every glove product was in Group 1, where TSI was less than 80%. Table 11 shows the prediction results of the high-performance models from the modeling state. Unexpectedly. The best model by LDA with second derivative spectral pretreatment + k-best wavelength selection could not predict accurately, but KNN developed by full spectra and Savitzky–Golay smoothing, + L2 norm scaling pretreatment, + GA wavelength selection spectra provided 100% accuracy for both one-layer and two-layer scans, but using the data after SMOTE (Table 12). It was observed that the results of most of the models show similar accuracy when scanned by both scan layers, inciting no prediction problem for the one-layer model by scanning a double-layer glove.

Figure 6 shows the raw spectra (Figure 6a) and the pretreated spectra by Savitzky–Golay smoothing + L2 norm scaling (Figure 6b) of the gloves in unknown samples, both one-layer and two-layer scans. The raw spectra show the same peaks as the raw spectra of the modeling set, but the peaks between 1400 and 1500 nm were shifted slightly to the left due to the slightly different production processes of the factories from ours. There were baseline shifts and tilling effects due to physical factors such as density, while the same peaks illustrated the same constituents. After being pretreated, the baseline effect was mostly eliminated, and the height of the peaks could inform the different radiation absorptions due to the different quantities of constituents.

## 4. Discussion

### 4.1. NIR Spectra of Medical Glove Samples Measured by MicroNIR Spectrometer

From Figure 3, the raw, smoothed, MSC, SNV, detrending, log transform, and L2 norm scaling spectra where the structure of the spectra was the same show obvious peaks of cis-1,4-polyisoprene from pure Para rubber sheet at 1200, 1390, and 1420 nm, as indicated by Sirisomboon et al. [5]. The MSC, SNV, detrending, and L2 norm scaling pretreated algorithms reduced the problem of baseline shift and tilling while the log transform could not, and the mezzy structure of the pretreated spectra was obtained from min-max scaling and RNV. The different TSI-level spectra were not clearly separated.

### 4.2. Statistics of Toluene Swell Index of Thick Film Samples for Modelling

Since the collected data are unbalanced (Table 3), applying SMOTE could help increase the sample size and balance the new data. This technique ensures an equal number of samples in each group or class, addressing the initial data imbalance. However, in our case, due to the spectral pretreatment effect on the prediction of the new data set obtained by SMOTE, the comparison of the before SMOTE models could not be compared to the after SMOTE models.

### 4.3. Performance of Classification Models

#### 4.3.1. Full Spectra

Amirruddin et al. [18] discussed the classification model’s ability to categorize accuracy as follows: less than 40.00% as poor, 40.00–80.00% as moderate, and more than 80.00% as good. When comparing the validation accuracy results for classifying three different TSI levels of thick-film medical gloves with an imbalanced dataset using the full spectrum range (Table 4), it was observed that ANN was an underfitting model, while SVM and kNN were considered moderate models, and LDA performed poorly.

The research data’s TSI level classification was also attempted using partial least squares regression (PLSR) with a method employed by Phanomsophon et al. [17]. When combined with various spectral pre-treatment algorithms, the classification accuracy ranged from 32.14% to 65.12%. This suggests a relative lower performance for PLSR in this context.

#### 4.3.2. Selective Wavelengths by k-Best and GA Method

The results of classifying three different TSI levels of thick-film medical gloves with an imbalanced dataset while using selected wavelengths through k-Best feature selection indicated that the removal of unimportant wavelengths could help mitigate overfitting issues (Table 6 and Table 7).

By comparison between the results in Table 6 and Table 7, though there were different spectral pretreatment methods, the models mostly improved due to SMOTE. However, by the guidelines of Amirruddin et al. [16], the model was still classified as having moderate performance.

When comparing the algorithms used for modeling, it was found that LDA developed from 2nd derivative preprocessing spectra with 15 k-Best selected wavelengths before SMOTE (Table 6) fairly accurately predicted the class but was most reliable among other algorithms, i.e., ANN, SVM, and kNN, due to higher prediction accuracy, precision, recall, and F1-score of the same value of 0.76 and no overfitting or underfitting prediction. The 15 featured wavelengths were in 4 ranges, including 930–960, 1335–1350, 1400–1500, and 1530–1600 nm, which were the vibrations of CH and CH_2_; CH_3_; CH, CH_2_, CONH_2_ and CONHR; and RNH_2_, respectively [32]. The model from the kNN algorithm combined with Savitzky–Golay smoothing + L2 norm scaling pretreated spectra, 63 GA wavelength selected, and SMOTE provided the best performance for unknown sample prediction. This model had an accuracy, precision, recall, and F1-score of 0.75 for the calibration set and 0.72, 0.73, 0.72, and 0.73, respectively, for the validation set. The 63 wavelengths were in 3 ranges, including 900–970 nm (CH 3rd overtone and OH 2nd overtone), 1000–1205 nm (NH 2nd overtone and CH 2nd overtone), and 1230–1660 nm (CH 2nd overtone, 1st overtone of CH combination, NH 1st overtone and CH 1st overtone) [32].

#### 4.3.3. Dimensional Reduction by PCA

The performance of the calibration model improved after applying SMOTE, especially with ANN and SVM. However, overfitting occurred during predictions on the prediction set (Table 9).

The best results were obtained with the LDA algorithm both before and after SMOTE, and these results were comparable to the LDA model with 15 k-Best selected wavelengths but without SMOTE. Therefore, the LDA model with 15 selected wavelengths had a shorter calculation time.

### 4.4. Effect of Sample Number

According to the principle that the model performance will be better if the number of samples is large due to the small error, in the case of ANN, the number of data values used for training must exceed the number of weights determined in the network; this entails using a large number of samples for calibration if the number of input variables is also large. Based on the results, PC-ANN, where the data dimension was reduced, was the best choice for the intended application [33]. In our case, we used GA-ANN, k-Best, and PC-ANN, where the variables were reduced from the original full spectra to some featured wavelength data. Recently, in 2023, Rasooli Sharabiani et al. [34] used ANN with samples of winter wheat leaf for evaluation of chlorophyll content based on VIS/NIR spectroscopy using PLSR and ANN, where 120 samples were for the training set and the left was for the test set. The models resulted in the most accurate predictions, with a correlation coefficient of 0.92 and 0.97, along with a root mean square error of 0.9131 and 0.7305, respectively. Ni et al. [35] suggested that back propagation ANN (BANN) were powerful and promising methods for handling linear as well as nonlinear systems, even when the data sets are moderately small, and they indicated that when very little data are available, BANN has the additional advantage of achieving robust predictive performance based on relatively small data sets compared to other nonlinear approaches while being less influenced by preprocessing, i.e., SNV.

### 4.5. Effect of SMOTE

SMOTE was utilized to address the issue of imbalanced samples within the class classifications, which were causing bias in classifying the smaller groups. We have used SMOTE in our synthetic oversampling to increase the number of data points to (42:42:42) from (42:29:22). Even in balanced data, the discrimination model’s performance does not significantly increase, indicating the inherent characteristics of the data set, in which there was a moderate correlation between spectral characteristics and the TSI of the medical glove samples. Generally, the discrimination performance of the model relatively increases before and after balancing the data for full spectra, selective spectra, and dimensional reduction spectra (See Table 11). Most of the models developed after SMOTE had a better F1-score. It shows that the SMOTE method can increase the discriminative model’s ability by generating similar but not identical data [36].

Table 11 shows the F1-scores of models for classifying TSI levels of medical gloves before and after SMOTE using no spectral pretreatment for each class. Specifically, the F1-scores for the TSI = 80–88% group and the TSI > 88% group increased after applying SMOTE. Therefore, SMOTE can address the bias in classifying smaller groups. In the best model constructed using LDA with 2nd derivative spectra and 15 k-best selected features, without employing SMOTE, the validation F1-score was slightly higher than by using SMOTE. However, the calibration F1-score without SMOTE is slightly lower than with SMOTE. Upon comparison, we opted for a model that does not utilize SMOTE. However, in other models, using SMOTE increases the F1 values of minorities.

In addition, the effect of SMOTE on increasing the model performance was remarkably confirmed by the 100% prediction accuracy of the unknown samples by the kNN algorithm combined with Savitzky–Golay smoothing + L2 norm scaling pretreated spectra, both in the full wavelength range and the 63 GA wavelength selected by the data after SMOTE (Table 10).

### 4.6. The Merit of This Study

We have tried in this work with different modeling methods, including wavelength selection and using balanced data, but the model performance has not improved. The machine learning algorithms in this study were from two linear classifier algorithms (LDA and kNN) and two hybrid classifier algorithms (linear and nonlinear), including SVM and ANN. We concluded that the inherent characteristics of the data set might explain the weak correlation between spectral characteristics and the TSI of the medical glove samples. Before we end everything, let us recall the chemistry of the biopolymers, in our case, natural rubber, and their relation to NIR radiation.

Natural rubber is a naturally occurring nanocomposite with an island-nanomatrix structure, which is composed of *cis*-1,4-polyisoprene particles with an average diameter of ~1 μm dispersed in a nanomatrix (several tens of nanometers thick) of nonrubber components such as proteins and phospholipids. The island-nanomatrix structure is stabilized by physical and chemical pinning with proteins and phospholipids, which is based on the fact that the *cis*-1,4-polyisoprene of natural rubber is a branched polymer [37]. A medical glove is a vulcanized natural rubber product where crosslinks must be formed between the polymer chains to provide adequate mechanical resistance for natural rubber latex products [38]. It was proved that *cis*-1,4-polyisoprene absorbed NIR radiation at 750, 907, and 920 nm in the NIR shortwavelength by Sirisomboon et al. [39] and at 1202, 1390, 1420, 1719, 1780, 1884, 2032, and 2218 nm in the NIR longwavelength by Sirisomboon et al. [5], and these absorptions correlate well with the chemical constituents in rubber latex, for example, dry rubber content [5,39,40] and total solids content [39,40] and ammonia [41] and with physical parameters such as viscosity [42].

The studies by Lim and Sirisomboon of crosslink density of natural rubber film developed from prevulcanized latex model which was created by PLSR using the spectra scanned by FT-NIR spectrometer [3], the natural rubber thin film model provided the r^2^, root mean square error of cross validation and bias of 0.65, 4.01%TSI and −0.028%TSI, respectively, using the wavenumber range of 6102–5446.3 cm^−1^ and 4428–4242.9 cm^−1^ (1639–1836 nm (included natural rubber absorption bands) and 2258–2357 nm), whereas for the natural rubber thick film model the r^2^, root mean square error of cross validation and bias were 0.70, 4.00%TSI and −0.006%TSI, respectively, using the wavenumber range of 6102–4597.7 cm^−1^ (1639–2175 nm (included natural rubber absorption bands)). By using the low-cost VIS/NIR diode array spectrometer in the wavelength range of 450–1000 nm using a fiber optic probe scanned on the thin and thick film, the models for crosslink density indicated by prevulcanisate relaxed modulus (PRM) had poor results, as indicated by the R^2^ of calibration and RMSEC of 0.02 and 17.31 × 10^4^ N/m^2^ and 0.05 and 16.63 × 10^4^ N/m^2^, respectively [43].

It is proved in our experiment in this report that the longer wavelength range between 960 and 1650 nm, including 3 bands of natural rubber absorption by a low-cost linear variable filter ultra-compact spectrometer (MicroNIR Pro, 1700ES, Viavi, USA), improved the NIR spectroscopy model by using LDA and kNN algorithms for classification with an accuracy of 76% for the validation set and 100% for the unknown set, respectively. Amirruddin et al. [44] categorized balanced accuracy as poor below 40.00%, moderate within 40.00–80.00%, and excellent above 80.00%. With 76% accuracy, it is obvious that the best model, constructed before SMOTE using LDA with 2nd derivative spectra and 15 k-Best selected features, had moderate model performance, and the kNN combined with Savitzky–Golay smoothing + L2 norm scaling pretreated spectra after SMOTE was excellent with 100% accuracy after being tested for 16 unknown samples, which were from only G1 (<80%TSI). This kNN model must be proved with more sample numbers and more sample groups. Therefore, both developed models can be implemented in the glove factory for screening purposes in the production line.

It might be concluded that this wavelength range spectra (960–1650), i.e., 15 wavelengths by the k-Best algorithm and 63 wavelengths by GA, contained crosslink density-featured information about the natural rubber-thick film of medical gloves.

In terms of speed of crosslink density measurement, the TSI reference laboratory test takes 15 min to get the result, and the sample is destroyed, while for non-destructive NIR spectroscopy, only 30s are needed. The NIR spectroscopy is suitable for homogeneous material where an NIR hyperspectral image is not necessary. The medical glove is a fairly homogeneous material; therefore, there is no need to use the hyperspectral image technique, which has a tremendously higher price.

## 5. Conclusions

Classification of the crosslink density level of para rubber thick film of medical gloves by using NIR spectral data combined with machine learning is the first time reported in this paper. The spectra were acquired by an ultra-compact portable MicroNIR spectrometer with a wavelength range of 960–1700 nm. The crosslink density levels were referencely evaluated by toluene swell index (TSI) to correlate with the NIR spectra of the medical glove samples generated using different levels of vulcanized compounded natural rubber latex, where the 3 groups of TSI were specified, including less than 80%TSI, 80–88%TSI, and more than 88%TSI. The 80–88%TSI group was the group in which the compounded latex was suitable for medical glove production, which made the glove specification comply with the required standards of customers, for example, ASTM standard and European standard.

Followed the ASTM standard for the glove when sold to the USA. By tensile testing, the torn point of the glove sample before and after accelerated curing must be more than or equal to 18 MPa with 650% elongation and more than or equal to 14 MPa with 500% elongation, respectively. By European glove trading before and after the torn point, the force must be more than or equal to 12 N and 9 N, respectively. The before-accelerated curing is 24 h after production. The accelerated curing is to simulate the storage or transportation in a container, in which the accelerating curing condition is when the glove sample is at 100 ± 2 °C for 22 ± 2 h or 70 ± 2 °C for 166 ± 2 h and kept at room temperature for not less than 1 h but not after 24 h and subjected to a tensile test.

From the experiment, we can conclude that when comparing the algorithms used for modeling, the LDA developed by 2nd derivative spectra using 15 k-Best selected wavelengths before SMOTE fairly accurately predicted the class, and the kNN combined with Savitzky–Golay smoothing + L2 norm scaling pretreated spectra after SMOTE showed an excellent accuracy of 100% on a 16/unknown sample set. But they were most reliable among other algorithms, i.e., ANN and SVM, due to higher prediction accuracy, precision, recall, and F1-scores of the same value of 0.76 and 1.00, respectively, with no overfitting or underfitting prediction. This developed model should be applied to factory usage for screening purposes. However, in deep learning modeling, for example, different types of convolutional neural networks should be explored to get a more accurate and reliable model with the larger sample number required for deep learning.

## Figures and Tables

**Figure 1 polymers-16-00184-f001:**
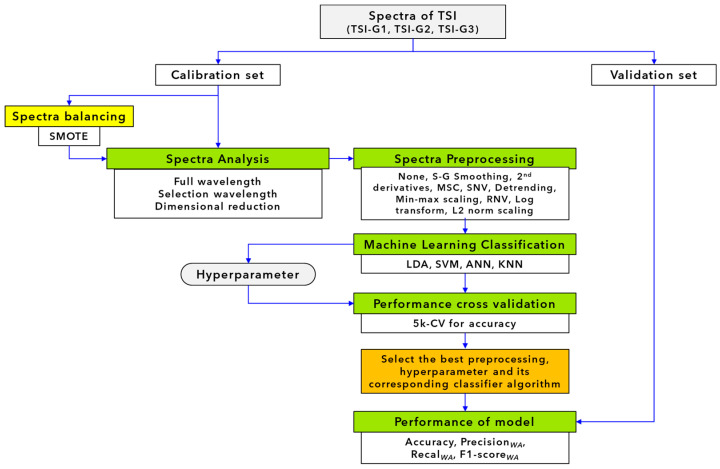
Flow diagram for model development of TSI classification.

**Figure 2 polymers-16-00184-f002:**
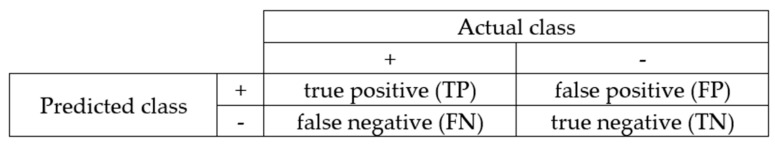
Confusion matrix.

**Figure 3 polymers-16-00184-f003:**
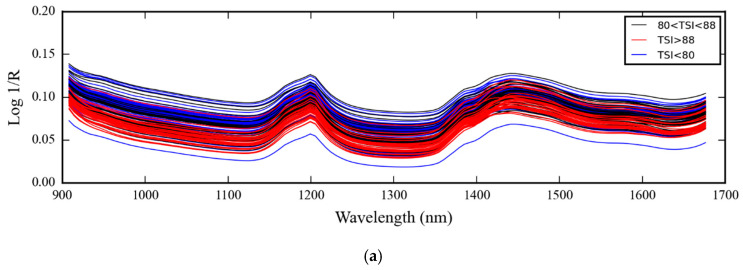
The (**a**) raw; (**b**) Savitzky–Golay smoothing; (**c**) 2nd derivative; (**d**) multiplicative scatter correction (MSC); (**e**) standard normal variate (SNV); (**f**) detrending; (**g**) min-max scaling; (**h**) robust normal variate (RNV); (**i**) log transform; (**j**) L2 norm scaling spectra of different levels of toluene swell index (Blue TSI less than 80%, Black TSI 80–88%, and Red TSI more than 88%).

**Figure 4 polymers-16-00184-f004:**
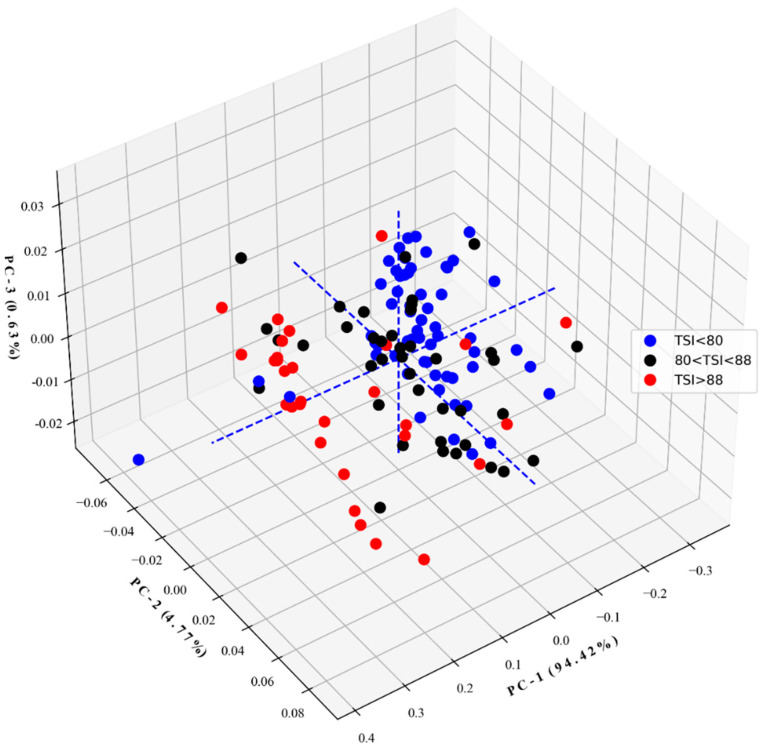
The 3D score plot from the full spectra of thick film samples.

**Figure 5 polymers-16-00184-f005:**
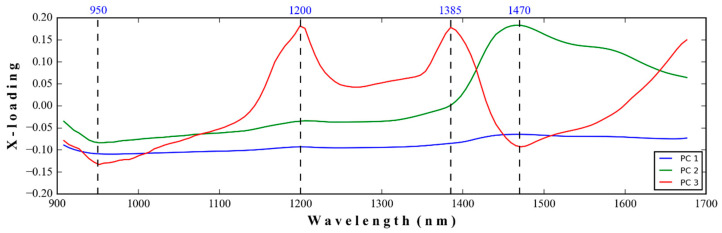
The first three X-loading lines are from full spectra of thick film samples.

**Figure 6 polymers-16-00184-f006:**
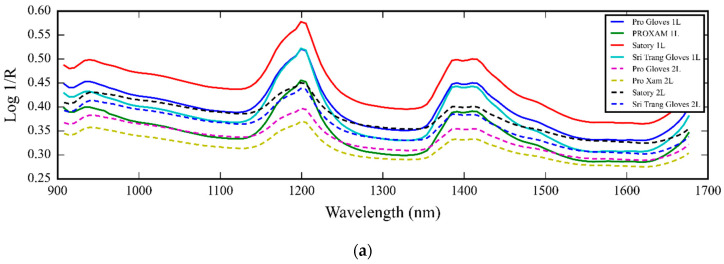
The (**a**) raw spectra and (**b**) pretreated spectra obtained by Savitzky–Golay smoothing + L2 norm scaling of the glove unknown samples were both one-layer and two-layer scans.

**Table 1 polymers-16-00184-t001:** Hyperparameter and tuning range of machine learning.

Algorithm	Hyperparameter	Range of Tuning
ANN	Hidden layer size (HLZ)	(3), (4), (5), (10), (11), (12), (16), (19), (20), (100), (3, 2), (5, 4), (100, 100), (4, 3, 2), (100, 100, 100)
	Activation function (AF)	identity, logistic, tanh, relu
SVM	Pinalty factor (C)	1–50
	Degree (D)	2, 3, 4
	Gamma (G)	scale, auto
kNN	n-neighbor (n)	1–20
LDA	n-component	1–20

**Table 2 polymers-16-00184-t002:** The model performance determination.

Parameter	Meaning	Formula
Accuracy	the proportion of correct predictions	$\frac{TP\;+\;TN}{N}$
Precision	Correctly predicting positive outcomes when the model predicts them as positive.	$\frac{TP}{TP\;+\;FP}$
Recall	the model’s capability of predicting positive cases	$\frac{TP}{TP\;+\;FN}$
F1-score	the harmonic means between precision and sensitivity	$\frac{Precision\;\times \;Recall}{Precision\;+\;Recall}$

The data analysis for this study was conducted utilizing MATLAB R2022b [29].

**Table 3 polymers-16-00184-t003:** The number of samples and statistics of TSI of thick film of medical glove samples of different levels for model development before and after SMOTE.

TSI Range (%)	Calibration Set					Prediction Set			
	Number of Samples before SMOTE	Number of Samples after SMOTE	True TSI Range (%)	Mean (%)	SD (%)	Number of Samples	True TSI Range (%)	Mean (%)	SD (%)
less than 80	42	42	71.00–79.17	75.42	2.22	17	72.92–79.17	75.90	2.01
80–88	29	42	80.00–88.00	83.39	2.67	12	81.25–88.00	84.42	2.49
more than 88	22	42	89.58–108.00	92.46	3.92	8	90.63–98.96	93.20	3.03

**Table 4 polymers-16-00184-t004:** Model performance for classifying TSI levels of thick film from medical gloves before SMOTE using full spectra.

Algorithm (WL)	Pre-Treatment	Hyper-Parameter	Calibration Set				Prediction Set			
			Weighted Average			A	Weighted Average			A
			P	R	F1		P	R	F1	
ANN (125)	Savitzky–Golay smoothing + RNV	HLZ = (100, 100, 100) AF = Identity	0.70	0.70	0.69	0.70	0.84	0.84	0.83	0.84
**SVM** **(125)**	**Savitzky–Golay smoothing** **+ RNV**	**C = 1** **D = 2** **G = scale**	**0.75**	**0.74**	**0.74**	**0.74**	**0.70**	**0.70**	**0.70**	**0.70**
kNN (125)	Savitzky–Golay smoothing + log transform	n = 6	0.77	0.77	0.76	0.77	0.69	0.70	0.69	0.70
LDA (125)	Second derivative	n = 1	0.99	0.99	0.99	0.99	0.49	0.46	0.44	0.46

P, Precision; R, Recall; F1, F1-score; A, Accuracy. The bold indicates the optimum model.

**Table 5 polymers-16-00184-t005:** Model performance for classifying TSI levels of thick film from medical gloves after SMOTE using full spectra.

Algorithm (WL)	Pre-Treatment	Hyper-Parameter	Calibration Set				Prediction Set			
			Weighted Average			A	Weighted Average			A
			P	R	F1		P	R	F1	
ANN (125)	Min-max normalization	HLZ = (100) AF = logistic	0.78	0.78	0.78	0.78	0.74	0.73	0.73	0.73
SVM (125)	Savitzky–Golay smoothing + L2 norm scaling	C = 3 D = 2 G = Scale	0.78	0.78	0.78	0.78	0.74	0.73	0.73	0.73
**kNN** **(125)**	**Savitzky–Golay smoothing + L2 norm scaling**	**n = 13**	**0.76**	**0.75**	**0.75**	**0.75**	**0.73**	**0.73**	**0.73**	**0.73**
LDA (125)	Savitzky–Golay smoothing + L2 norm scaling	n = 1	0.99	0.99	0.99	0.99	0.45	0.43	0.43	0.43

P, Precision; R, Recall; F1, F1-score; A, Accuracy. The bold indicates the optimum model.

**Table 6 polymers-16-00184-t006:** Model performance for classifying TSI levels of thick film from medical gloves before SMOTE using selective spectra.

Selection Method	Algorithm (WL)	Pre-Treatment	Hyper-Parameter	Calibration Set				Prediction Set			
				Weighted Average			A	Weighted Average			A
				P	R	F1		P	R	F1	
k-Best	ANN (41)	Savitzky–Golay smoothing + RNV	HLZ = (100) AF = relu k-best = f_classif	0.74	0.74	0.74	0.74	0.67	0.68	0.66	0.68
	SVM (14)	Savitzky–Golay smoothing + RNV	C = 27 D = 2 G = scale k-best = mutual_info_classif	0.79	0.78	0.78	0.78	0.64	0.65	0.63	0.65
	kNN (19)	Savitzky–Golay smoothing + log transform	n = 4 k-best = mutual_info_classif	0.75	0.75	0.74	0.75	0.67	0.68	0.65	0.68
	**LDA** **(15)**	**Second derivative**	**n = 1** **k-best = f_classif**	**0.72**	**0.72**	**0.72**	**0.72**	**0.76**	**0.76**	**0.76**	**0.76**
GA	ANN (60)	Savitzky–Golay smoothing + RNV	HLZ = (10) AF = relu	0.71	0.71	0.70	0.71	0.66	0.65	0.63	0.65
	SVM (55)	Savitzky–Golay smoothing + RNV	C = 1 D = 2 G = scale	0.75	0.74	0.74	0.74	0.70	0.70	0.70	0.70
	kNN (62)	Savitzky–Golay smoothing + log transform	n = 6	0.76	0.76	0.75	0.76	0.69	0.70	0.69	0.70
	LDA (60)	Second derivative	n = 1	0.91	0.91	0.91	0.91	0.64	0.65	0.64	0.65

P, Precision; R, Recall; F1, F1-score; A, Accuracy. The bold indicates the optimum model.

**Table 7 polymers-16-00184-t007:** Model performance for classifying TSI levels of thick film from medical gloves after SMOTE using selective spectra.

Selection Method	Algorithm (WL)	Pre-Treatment	Hyper-Parameter	Calibration Set				Prediction Set			
				Weighted Average			A	Weighted Average			A
				P	R	F1		P	R	F1	
k-best	ANN (22)	Min-max scaling	HLZ = (100, 100) AF = relu k-best = chi2	0.75	0.75	0.74	0.75	0.70	0.70	0.70	0.70
	SVM (43)	Savitzky–Golay smoothing + L2 norm scaling	C = 1, D = 2 G = Scale k-best = mutual_info_classif	0.80	0.79	0.79	0.79	0.68	0.65	0.66	0.65
	kNN (1)	Savitzky–Golay smoothing + L2 norm scaling	n = 7 k-best = chi2	0.80	0.79	0.79	0.80	0.68	0.65	0.66	0.68
	LDA (2)	Savitzky–Golay smoothing + L2 norm scaling	n = 1 k-best = mutual_info_classif	0.75	0.75	0.75	0.75	0.70	0.70	0.70	0.70
GA	ANN (61)	Min-max scaling	HLZ = (100, 100) AF = relu	0.80	0.80	0.80	0.80	0.74	0.73	0.73	0.73
	SVM (69)	Savitzky–Golay smoothing + L2 norm scaling	C = 2, D = 2 G = Scale	0.78	0.78	0.78	0.78	0.73	0.70	0.71	0.70
	**kNN** **(63)**	**Savitzky–Golay smoothing + L2 norm scaling**	**n = 19**	**0.75**	**0.75**	**0.75**	**0.75**	**0.72**	**0.73**	**0.72**	**0.73**
	LDA (74)	Savitzky–Golay smoothing + L2 norm scaling	n = 1	0.99	0.99	0.99	0.99	0.57	0.57	0.57	0.57

P, Precision; R, Recall; F1, F1-score; A, Accuracy. The bold indicates the optimum model.

**Table 8 polymers-16-00184-t008:** Model performance for classifying TSI levels of thick film from medical gloves before SMOTE using dimensional reduction by PCA.

Algorithm	Pretreatment	Hyper- Parameter	Calibration Set				Prediction Set			
			Weighted Average			A	Weighted Average			A
			P	R	F1		P	R	F1	
10PC-ANN	Savitzky–Golay smoothing + Mean scaling	HLZ = (100) AF = relu	0.74	0.74	0.74	0.74	0.75	0.76	0.75	0.76
10PC-SVM	Savitzky–Golay smoothing	C = 1 D = 2 G = scale	0.75	0.74	0.74	0.74	0.70	0.70	0.70	0.70
10PC-kNN	Savitzky–Golay smoothing + Log transform	n = 6	0.77	0.77	0.76	0.77	0.69	0.70	0.69	0.70
**10PC-LDA**	**Savitzky–Golay smoothing + Detrending**	**n = 1**	**0.76**	**0.76**	**0.76**	**0.76**	**0.75**	**0.76**	**0.75**	**0.75**

P, Precision; R, Recall; F1, F1-score; A, Accuracy. The bold indicates the optimum model.

**Table 9 polymers-16-00184-t009:** Model performance for classifying TSI levels of thick film from medical gloves after SMOTE using dimensional reduction by PCA.

Algorithm	Pretreatment	Hyper- Parameter	Calibration Set				Prediction Set			
			Weighted Average			A	Weighted Average			A
			P	R	F1		P	R	F1	
10PC-ANN	Savitzky–Golay smoothing + RNV	HLZ = (100, 100, 100) AF = relu	0.94	0.94	0.94	0.94	0.61	0.62	0.61	0.62
10PC-SVM	Second derivative	C = 25 D = 2 G = Scale	0.98	0.98	0.98	0.98	0.70	0.68	0.68	0.68
10PC-kNN	Savitzky–Golay smoothing + Log transform	n = 3	0.83	0.83	0.83	0.83	0.66	0.68	0.65	0.68
**10PC-LDA**	**Savitzky–Golay smoothing + Robust normal variate**	**n = 1**	**0.76**	**0.76**	**0.76**	**0.76**	**0.75**	**0.76**	**0.75**	**0.76**

P, Precision; R, Recall; F1, F1-score; A, Accuracy. The bold indicates the optimum model.

**Table 10 polymers-16-00184-t010:** The information about an unknown medical glove product from different factories.

Factory	Production Date	Expired Date	Initial Diameter (mm)	% TSI
sl1	August 2022	August 2025	40	60
sl2	August 2022	August 2025	40	60
sl3	August 2022	August 2025	40	60
sl4	August 2022	August 2025	40	60
PO1	May 2023	May 2026	40	60
PO2	May 2023	May 2026	40	60
PO3	May 2023	May 2026	40	60
PO4	May 2023	May 2026	40	60
PX 1	October 2023	October 2026	40	60
PX 2	October 2023	October 2026	40	60
PX 3	October 2023	October 2026	40	60
PX 4	October 2023	October 2026	40	60
St1	September 2023	September 2026	43	72
St2	September 2023	September 2026	40	60
St3	September 2023	September 2026	40	60
St4	September 2023	September 2026	40	60

**Table 11 polymers-16-00184-t011:** Accuracy of classification of the TSI level of unknown samples by developed models using before and after SMOTE data.

Full Spectrum	Best Preprocessing + Algorithm	Scanning Method	Group		
			G1 (TSI < 80)	G2 (80 < TSI < 88)	G3 (TSI > 88)
Before SMOTE	(Savitzky–Golay smoothing + RNV) + SVM	One-layer scan	288	12	20
		Two-layer scan	276	21	23
**After SMOTE**	**(Savitzky–Golay smoothing + L2 norm scaling) + kNN**	**One-layer scan**	**320**	**0**	**0**
		**Two-layer scan**	**320**	**0**	**0**
**Selection Wavelength**	**Best Preprocessing + Algorithm**	**Scanning Method**	**Group**		
			**G1 (TSI < 80)**	**G2 (80 < TSI < 88)**	**G3 (TSI > 88)**
Before SMOTE	(Second derivative) + k-best + LDA	One-layer scan	55	1	264
		Two-layer scan	263	45	12
**After SMOTE**	**(Savitzky–Golay smoothing + L2 norm scaling) + GA + kNN**	**One-layer scan**	**320**	**0**	**0**
		**Two-layer scan**	**320**	**0**	**0**
**Reduction Features**	**Best Preprocessing + Algorithm**	**Scanning Method**	**Group**		
			**G1 (TSI < 80)**	**G2 (80 < TSI < 88)**	**G3 (TSI > 88)**
Before SMOTE	(Savitzky–Golay smoothing + Detrending) + 10-PC + LDA	One-layer scan	128	104	88
		Two-layer scan	137	104	79
After SMOTE	(Savitzky–Golay smoothing + RNV) + 10-PC + LDA	One-layer scan	141	69	110
		Two-layer scan	127	92	101

The bold indicates the optimum model.

**Table 12 polymers-16-00184-t012:** The F1-score of models for classifying TSI levels of medical gloves before and after SMOTE using no spectral pretreatment.

Algorithm	Selection Method	Calibration						Validation					
		Before SMOTE			After SMOTE			Before SMOTE			After SMOTE		
		<80%	80–88%	>80%	<80%	80–88%	>80%	<80%	80–88%	>80%	<80%	80–88%	>80%
ANN	Full	0.78	0.62	0.63	0.80	0.75	0.78	0.89	0.73	0.88	0.79	0.62	0.80
	k-best	0.83	0.66	0.67	0.79	0.68	0.77	0.79	0.45	0.71	0.78	0.52	0.88
	GA	0.86	0.65	0.46	0.86	0.75	0.80	0.83	0.52	0.36	0.79	0.62	0.80
SVM	Full	0.81	0.69	0.65	0.83	0.74	0.77	0.81	0.52	0.71	0.79	0.62	0.80
	k-best	0.84	0.74	0.72	0.82	0.76	0.78	0.77	0.38	0.71	0.73	0.52	0.71
	GA	0.81	0.69	0.65	0.83	0.74	0.77	0.81	0.52	0.71	0.71	0.67	0.80
kNN	Full	0.87	0.70	0.65	0.82	0.70	0.74	0.82	0.48	0.71	0.80	0.58	0.80
	k-best	0.83	0.68	0.67	0.81	0.78	0.78	0.78	0.42	0.71	0.73	0.52	0.71
	GA	0.86	0.68	0.65	0.82	0.70	0.74	0.82	0.48	0.71	0.81	0.55	0.75
LDA	Full	1.00	0.98	0.98	1.00	0.99	0.99	0.50	0.29	0.56	0.47	0.36	0.45
	k-best	0.81	0.63	0.67	0.81	0.68	0.77	0.79	0.64	0.88	0.78	0.52	0.80
	GA	0.95	0.88	0.88	1.00	0.99	0.99	0.73	0.45	0.74	0.69	0.61	0.25

## Data Availability

Data are contained within the article.

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
