# Peer review of "Classification of the Crosslink Density Level of Para Rubber Thick Film of Medical Glove by Using Near-Infrared Spectral Data"

_polymers, 2024, doi:10.3390/polym16020184_

Round 1
Reviewer 1 Report
Comments and Suggestions for Authors
Although this study showed that near-infrared spectroscopy can be used to classify the crosslink density level of medical gloves, the performance of the developed models still needs to be improved. Increasing the number of samples, balancing the data, and validating in real conditions can help improve the performance of the model.
1. The total number of samples used in this study was only 155, of which 93 were used for modeling. This number is too small for deep learning algorithms like convolutional neural networks, leading to poorer performance. Increasing the number of samples is essential.
2. The sample ratio in the 3 classification groups was imbalanced (42:29:22), causing bias in classifying the smaller groups. The SMOTE method for balancing was ineffective.
3. SMOTE preprocessing introduced noise in the spectra and reduced the predictive ability of the models. Other methods like undersampling the dominant group could be utilized.
4. The best classification accuracy was around 76% indicating moderate model performance. Improving algorithms and increasing samples could help increase accuracy.
5. The models were only trained on laboratory data. Testing the models in real manufacturing conditions is crucial to evaluate their efficacy.
6. The quality of the images should be improved and technically should be authors informed of the difference between the methods that authors considered for this paper.
Overall, the quality and quantity of data and the algorithms used need improvement to develop models with higher accuracy and reliability.
Author Response
Reviewer 1
Although this study showed that near-infrared spectroscopy can be used to classify the crosslink density level of medical gloves, the performance of the developed models still needs to be improved. Increasing the number of samples, balancing the data, and validating in real conditions can help improve the performance of the model.
- The total number of samples used in this study was only 155, of which 93 were used for modeling. This number is too small for deep learning algorithms like convolutional neural networks, leading to poorer performance. Increasing the number of samples is essential.
Thank you for your suggestion; we value your advice regarding the importance of increasing the sample size for deep learning algorithms.
We would like to explain that we have not used deep learning like CNN which a huge number of data is necessary. Also, in this study, we have considered 2 algorithms as linear classifiers (LDA, KNN) and 2 algorithms as hybrid classifiers (linear and nonlinear ), including SVM and ANN, with the total real number of spectra in this study is 130, of which 93 (42:29:22) were used as training and 37 as testing. With the SMOTE method, the total spectra were increased to 163, of which 126 (42:42:42) were used as training, and 37 real spectra were used as testing. The samples collection duration of our experiment was during 28 January 2022 till 9 January 2023 which is the year-round production of the factory, though less in your opinion but this confirm the robustness with wide variation data of our calibration model.
However, it's important to note that sample collection is a time-consuming process. Within this study, we managed to collect 130 samples over the span of one year. Gathering additional samples presents a difficulty due to the significant time investment it demands.
The following is input in discussion in the small sample number issue in red color
4.4 Effect of sample number
By the principle that the model performance will be better , if the the number of samples is large due to the small error. In case of ANN, the number of data values used for training must exceed that of weights determined in the network; this entails using a large number of samples for calibration if the number of input variables is also large. Based on the results, PC-ANN, where the data dimension was reduced, was the best choice for the intended application [33], in which our case we used GA-ANN, k Best and PC-ANN where the variables was reduced from the original of full spectra to some featured wavelength data. Recently in 2023, Rasooli Sharabiani et al [34] used ANN with samples of winter wheat leaf for evaluation of chlorophyll content based on VIS/NIR spectroscopy using PLSR and ANN where 120 samples was for training set and the left was for test set and the models result the most accurate prediction, with the correlation coefficient of 0.92 and 0.97, along with the root mean square error of 0.9131 and 0.7305 respectively. Ni et al [35] suggested that back propagation ANN (BANN) were powerful and promising methods for handling linear as well as nonlinear systems, even when the data sets are moderately small and they indicated that when very little data is available BANN had an additional advantage of achieving robust predictive performance based on relatively small data sets compared to other nonlinear approaches while being less influenced by preprocessing, i.e. SNV.
[33] Blanco, M., Coello, J., Iturriaga, H., Maspoch, S., & Pages, J. (2000). NIR calibration in non-linear systems: different PLS approaches and artificial neural networks. Chemometrics and Intelligent Laboratory Systems, 50(1), 75-82.
[34] Rasooli Sharabiani, V., Soltani Nazarloo, A., Taghinezhad, E., Veza, I., Szumny, A., & Figiel, A. (2023). Prediction of winter wheat leaf chlorophyll content based on VIS/NIR spectroscopy using ANN and PLSR. Food Science & Nutrition, 11(5), 2166-2175.
[35] Ni, W., Nørgaard, L., & Mørup, M. (2014). Non-linear calibration models for near infrared spectroscopy. Analytica chimica acta, 813, 1-14.
- The sample ratio in the 3 classification groups was imbalanced (42:29:22), causing bias in classifying the smaller groups. The SMOTE method for balancing was ineffective.
Due to the numerous results, we regretfully did not present the F1-scores of calibrations for each level, resulting in unclear conclusions. Table 11 displays the F1-scores for each class is included now in revised manuscript.
The following is input in the discussion of revised manuscript where appropriate with red color
4.5 Effect of SMOTE
SMOTE was utilized to address the issue of imbalanced samples within the class classifications, which were causing bias in classifying the smaller groups. We have used SMOTE in our synthetic oversampling make number of data increased to (42:42:42) from (42:29:22). Even in balanced data, the discrimination model's performance relatively does not significantly increase, indicating the inherent characteristics of the data set in which there were moderate correlation between spectral characteristics and the TSI of the medical glove samples. Generally, the discrimination performance of the model relatively increases before and after balancing the data for full spectra, selective spectra, and dimensional reduction spectra (See Figure 4 to Figure 5). It shows that the SMOTE method can increase the discriminative model's ability by generating similar but not identical data [36].
Table 11 shows the F1-scores of models for classifying TSI levels of medical gloves before and after SMOTE using no spectral pretreatment for each class. Specifically, the F1-scores for the TSI = 80-88% group and TSI > 88% group increased after applying SMOTE. Therefore, SMOTE can address the bias in classifying smaller groups. In the best model constructed using LDA with 2nd derivative spectra and 15 k-best selected features, without employing SMOTE, the validation F1-score was slightly higher than by using SMOTE. However, the calibration F1-score without SMOTE is slightly lower than with SMOTE. Upon the comparison, we opted for a model that does not utilize SMOTE. However, in other models, using SMOTE increases the F1 values of minorities group.
Besides the effect of SMOTE to increase the model performance was remarkably confirmed by 100% prediction accuracy of the unknown samples by kNN algorithm combined with Savitzky-Golay smoothing + L2เ norm scaling pretreated spectra, both the full wavelength range and the 63 GA wavelength selected by the data after SMOTE (Table 10).
Table 11. The F1-score of models for classifying TSI levels of medical gloves before and after SMOTE using no spectral pretreatment
|
Algorithm |
Selection method |
Calibration |
Validation |
||||||||||
|
Before SMOTE |
After SMOTE |
Before SMOTE |
After SMOTE |
||||||||||
|
< 80% |
80-88 % |
> 80% |
< 80% |
80-88 % |
> 80% |
< 80% |
80-88 % |
> 80% |
< 80% |
80-88 % |
> 80% |
||
|
ANN |
Full |
0.78 |
0.62 |
0.63 |
0.80 |
0.75 |
0.78 |
0.89 |
0.73 |
0.88 |
0.79 |
0.62 |
0.80 |
|
k-best |
0.83 |
0.66 |
0.67 |
0.79 |
0.68 |
0.77 |
0.79 |
0.45 |
0.71 |
0.78 |
0.52 |
0.88 |
|
|
GA |
0.86 |
0.65 |
0.46 |
0.86 |
0.75 |
0.80 |
0.83 |
0.52 |
0.36 |
0.79 |
0.62 |
0.80 |
|
|
SVM |
Full |
0.81 |
0.69 |
0.65 |
0.83 |
0.74 |
0.77 |
0.81 |
0.52 |
0.71 |
0.79 |
0.62 |
0.80 |
|
k-best |
0.84 |
0.74 |
0.72 |
0.82 |
0.76 |
0.78 |
0.77 |
0.38 |
0.71 |
0.73 |
0.52 |
0.71 |
|
|
GA |
0.81 |
0.69 |
0.65 |
0.83 |
0.74 |
0.77 |
0.81 |
0.52 |
0.71 |
0.71 |
0.67 |
0.80 |
|
|
kNN |
Full |
0.87 |
0.70 |
0.65 |
0.82 |
0.70 |
0.74 |
0.82 |
0.48 |
0.71 |
0.80 |
0.58 |
0.80 |
|
k-best |
0.83 |
0.68 |
0.67 |
0.81 |
0.78 |
0.78 |
0.78 |
0.42 |
0.71 |
0.73 |
0.52 |
0.71 |
|
|
GA |
0.86 |
0.68 |
0.65 |
0.82 |
0.70 |
0.74 |
0.82 |
0.48 |
0.71 |
0.81 |
0.55 |
0.75 |
|
|
LDA |
Full |
1.00 |
0.98 |
0.98 |
1.00 |
0.99 |
0.99 |
0.50 |
0.29 |
0.56 |
0.47 |
0.36 |
0.45 |
|
k-best |
0.81 |
0.63 |
0.67 |
0.81 |
0.68 |
0.77 |
0.79 |
0.64 |
0.88 |
0.78 |
0.52 |
0.80 |
|
|
GA |
0.95 |
0.88 |
0.88 |
1.00 |
0.99 |
0.99 |
0.73 |
0.45 |
0.74 |
0.69 |
0.61 |
0.25 |
|
[36] Hao, Y., Li, X., & Zhang, C. (2023). Improving prediction model robustness with virtual sample construction for near-infrared spectra analysis. Analytica Chimica Acta, 1279, 341763.
- SMOTE preprocessing introduced noise in the spectra and reduced the predictive ability of the models. Other methods like undersampling the dominant group could be utilized.
Thank you for your comment. We would like to provide further insight into SMOTE.
We would like to explain that SMOTE cannot introduce noise in the spectra if the real spectra have no noise. In our case, most of spectra set the Savitzky-Golay smoothing was applied and if you observe Figure 3 a-j you can see no noise or less noise in the spectra
We then input the following in the methodology for the spectral pre-processing section in revised manuscript
SMOTE involves oversampling through the creation of synthetic new samples based on the original ones. These new samples are generated using random constants within the range of [0,1] and are placed at distances determined by the original samples [13]. As a result, SMOTE preprocessing typically does not introduce noise unless the original samples contain noise, in which case the new spectra might inherit noise. To address this issue, we employed Savitzky-Golay smoothing, which helped mitigate the problem.
[13] Chawla, N.V., et al., SMOTE: Synthetic Minority Over-sampling Technique. ournal of Artificial Intelligence Research, 2002. 16: p. 321-357.
Undersampling can be effective in addressing imbalanced groups. However, given out smaller sample size, Therefore, employing undersampling may not be suitable.
- The best classification accuracy was around 76% indicating moderate model performance. Improving algorithms and increasing samples could help increase accuracy.
Thank you so much for your suggestion. Increasing the sample size was difficult due to time constraints. This study aimed to develop models using various algorithms (ANN, SVM, kNN, and LDA) and different feature selection methods (k-Best and GA) to determine the optimal model for classifying the crosslink density level of para rubber thick film for medical gloves.
It is proved in our experiment in this report that the longer wavelength range between 960-1650 nm including 3 bands of natural rubber absorption by low cost linear variable filter ultra compact spectrometer (MicroNIR Pro, 1700ES, Viavi, USA) had improved the NIR spectroscopy model by using LDA and kNN algorithms for classification with accuracy of 76% for validation set and 100% for unknown set, respectively. Amirruddin et al. [44] categorized balanced accuracies as poor below 40.00%, moderate within 40.00%–80.00%, and excellent above 80.00%. With 76% accuracy, it is obvious that the best model, constructed before SMOTE using LDA with 2nd derivative spectra and 15 k-Best selected features was moderate model performance and the kNN combined with Savitzky-Golay smoothing + L2เ norm scaling pretreated spectra after SMOTE was excellent with 100% accuracy after tested for 16 unknown samples which was from only G1 (< 80%TSI). This kNN model must be proved with more sample number and more sample groups. Therefore, both developed models can be implemented in the glove factory for screening purpose in the production line.
[44] Amirruddin, A.D., et al., Hyperspectral remote sensing for assessment of chlorophyll sufficiency levels in mature oil palm (Elaeis guineensis) based on frond numbers: Analysis of decision tree and random forest. Computers and Electronics in Agriculture, 2020. 169: p. 105221.
- The models were only trained on laboratory data. Testing the models in real manufacturing conditions is crucial to evaluate their efficacy.
We then did experiment using 16 real product of medical gloves of 4 factories
2.6 Validation by unknown real sample set form factories
Medical rubber gloves without powder was collected from 4 factories (4 gloves for 1 factory). Therefore, 16 gloves. Each glove was scanned by placing MicroNIR window on the intact glove (two-layer scan) placed on the aluminum plate as a reflector make this scanning the transflectance mode. The scanning was done 5 scans per position and 4 positions on one glove. Therefore, 320 spectra in total. Then, the MicroNIR was inserted inside the glove and scanned the glove only one layer by the same procedure. After that the glove was subjected to the TSI test immediately.
Every spectrum scanned was subjected to some models developed including LDA, kNN and SVM with different pretreatment methods and the model classification performance were calculated.
3.1.4 Validation result by unknown real sample set form factories
Table 10 shows the production information of the unknown glove samples and the TSI value of every sample. The models developed were used to predict the TSI value of the glove sample. The TSI of every glove was 60% except one was 72% which was in group 1 (<80%). This indicated the uniformity of the gloves production. Though, there may have an opportunity to have out of accetable group indicating the need of non-destructive detection of the product real time on-line for 100% crossling density levels detection.
By TSI test, the result show that every glove product was in group 1 where TSI was less than 80%. Table 11 show the prediction results of the high performance models from modelling state. Unexpectedly. The best model by LDA with second derivative spectral pretreatment + k-best wavelength selection could not predict accurately but KNN developed by full spectra and Savitzky-Golay smoothing + L2 norm scaling pretreatment + GA wavelength selection spectra provided 100% accuracy for both scan on one-layer or two-layer scan but using the data after SMOTE (Table 11). It was observed that the results of most of the models show similar accuracy when scan by both scan-layer incating no prediction problem of one-layer model by scanning of double layer glove.
Figure 6 shows the raw spectra (Figure 6 (a)) and the pretreated spectra by Savitzky-Golay smoothing + L2 norm scaling (Figure 6 (b) of the gloves in unknown samples both one-layer and two-layer scan. The raw spectra shows the baseline shift and tillting effects due to physical factors such as density while the same peaks illustraed the same constituents. After pretreated, the baseline effect was mostly eliminated and the hight of peaks could inform the different radiation absorption due to different quantity of constituents.
Table 10. The information of unknown medical glove product from different factories
|
Factory |
production date |
expired date |
initial diameter (mm) |
% TSI |
|
sl1 |
08-2022 |
08-2025 |
40 |
60 |
|
sl2 |
08-2022 |
08-2025 |
40 |
60 |
|
sl3 |
08-2022 |
08-2025 |
40 |
60 |
|
sl4 |
08-2022 |
08-2025 |
40 |
60 |
|
PO1 |
05-2023 |
05-2026 |
40 |
60 |
|
PO2 |
05-2023 |
05-2026 |
40 |
60 |
|
PO3 |
05-2023 |
05-2026 |
40 |
60 |
|
PO4 |
05-2023 |
05-2026 |
40 |
60 |
|
PX 1 |
10-2023 |
10-2026 |
40 |
60 |
|
PX 2 |
10-2023 |
10-2026 |
40 |
60 |
|
PX 3 |
10-2023 |
10-2026 |
40 |
60 |
|
PX 4 |
10-2023 |
10-2026 |
40 |
60 |
|
St1 |
09-2023 |
09-2026 |
43 |
72 |
|
St2 |
09-2023 |
09-2026 |
40 |
60 |
|
St3 |
09-2023 |
09-2026 |
40 |
60 |
|
St4 |
09-2023 |
09-2026 |
40 |
60 |
Table 11. Accuracy of classification of toluene swell index level of unknown samples by developled models using the before SMOTE and after SMOTE.
|
Full Spectrum |
Best preprocessing + algorithm |
Scanning method |
Group |
||
|
G1 (TSI < 80) |
G2 (80<TSI<88) |
G3 (TSI > 88) |
|||
|
Before SMOTE |
(Savitzky-Golay smoothing + RNV) + SVM |
One-layer scan |
288 |
12 |
20 |
|
Two-layer scan |
276 |
21 |
23 |
||
|
After SMOTE |
(Savitzky-Golay smoothing + L2 norm scaling) + kNN |
One-layer scan |
320 |
0 |
0 |
|
Two-layer scan |
320 |
0 |
0 |
||
|
Selection wavelength |
Best preprocessing + algorithm |
Scanning method |
Group |
||
|
1 (TSI < 80) |
2 (80<TSI<88) |
3 (TSI > 88) |
|||
|
Before SMOTE |
(Second derivative) + k-best + LDA |
One-layer scan |
55 |
1 |
264 |
|
Two-layer scan |
263 |
45 |
12 |
||
|
After SMOTE |
(Savitzky-Golay smoothing + L2เ norm scaling) + GA + kNN |
One-layer scan |
320 |
0 |
0 |
|
Two-layer scan |
320 |
0 |
0 |
||
|
Reduction features |
Best preprocessing + algorithm |
Scanning method |
Group |
||
|
1 (TSI < 80) |
2 (80<TSI<88) |
3 (TSI > 88) |
|||
|
Before SMOTE |
(Savitzky-Golay smoothing + Detrending) + 10-PC + LDA |
One-layer scan |
128 |
104 |
88 |
|
Two-layer scan |
137 |
104 |
79 |
||
|
After SMOTE |
(Savitzky-Golay smoothing + RNV) + 10-PC + LDA |
One-layer scan |
141 |
69 |
110 |
|
Two-layer scan |
127 |
92 |
101 |
||
(a)
(b)
Figure 6 The (a) raw spectra and (b) the pretreated spectra by Savitzky-Golay smoothing + L2 norm scaling of the gloves in unknown samples both one-layer and two-layer scan.
- The quality of the images should be improved and technically should be authors informed of the difference between the methods that authors considered for this paper.
The images were improved to have a resolution of 300 dpi.
Overall, the quality and quantity of data and the algorithms used need improvement to develop models with higher accuracy and reliability.
Thank you so much for your suggestions.
In this study, we have already reported the model discrimination performance of several machine learning classifiers for full spectra, feature selection, and dimension reduction. We can group into a classifier that can work linearly (LDA, KNN) and a classifier that works hybrid (SVM, ANN).
Because the data is very challenging due to the weak relationship between the NIR spectrum and the response from the TSI of the medical glove samples, the performance of the machine learning classifier is poor.
We would like to explain that we have not used deep learning like CNN which a huge number of data is necessary. Also, in this study, we have considered 2 algorithms as linear classifiers (LDA, KNN) and 2 algorithms as hybrid classifiers (linear and nonlinear ), including SVM and ANN, with the total real number of spectra in this study is 130, of which 93 (42:29:22) were used as training and 37 as testing. With the SMOTE method, the total spectra were increased to 163, of which 126 (42:42:42) were used as training, and 37 real spectra were used as testing. The samples collection duration of our experiment was during 28 January 2022 till 9 January 2023 which is the year-round production of the factory, though less in your opinion but this confirm the robustness with wide variation data of our calibration model.
However, it's important to note that sample collection is a time-consuming process. Within this study, we managed to collect 130 samples over the span of one year. Gathering additional samples presents a difficulty due to the significant time investment it demands.
However, by unknown test explained in your recommendation 5 it is proved our develped model accurately workable.

Reviewer 2 Report
Comments and Suggestions for Authors
This paper describes for the first time the use of IR spectroscopy in combination with machine learning to classify the cross-link density level of para-rubber medical gloves. The model developed using this approach, in further, can be implemented in a glove factory for testing on the production line. It is worth noting, however, that deep learning modeling should be studied in more detail. Overall, this work can be potentially interesting for the readers of this journal. The manuscript itself is well written and presented. Thus, I recommend this manuscript to be published as it.
Author Response
Reviewer 2
This paper describes for the first time the use of IR spectroscopy in combination with machine learning to classify the cross-link density level of para-rubber medical gloves. The model developed using this approach, in further, can be implemented in a glove factory for testing on the production line. It is worth noting, however, that deep learning modeling should be studied in more detail. Overall, this work can be potentially interesting for the readers of this journal. The manuscript itself is well written and presented. Thus, I recommend this manuscript to be published as it.
Thanks so much for your recommendation to publish our manuscript in Polymers as it is.
Reviewer 3 Report
Comments and Suggestions for Authors
In this manuscript, authors demonstrate characterization of latex using near infrared spectroscopy combined with machine learning. While utilizing a non-destructive method like near infrared spectroscopy can be advantages and interesting, I suggest authors to make revisions to improve the manuscript.
1. It would be helpful to elaborate further on the advantages of using NIR spectroscopy technique. Perhaps add more references that supports the statements regarding its speed and accuracy. Also emphasize why NIR spectroscopy is appropriate for this study rather than other non-destructive methods that can be used.
2. The manuscript lacks detailed description of the machine learning algorithms used in this study. Provide brief description of the algorithms/methods with relevant references.
3. I suggest using different line types for different TSI in Figure 3 for better legibility.
4. Provide further explanation on why LDA achieved the highest accuracy during calibration however exhibited the lowest accuracy after validation.
5. Provide relevant references for the data processing methods used in this study, for example Savitzky – Golay smoothing.
Comments on the Quality of English LanguageThe overall quality of English in this manuscript is very poor; current manuscript contains a lot of grammatical errors and typos, which makes it difficult to follow. The authors must carefully review and revise the entire manuscript for better English.
Author Response
Reviewer 3
In this manuscript, authors demonstrate characterization of latex using near infrared spectroscopy combined with machine learning. While utilizing a non-destructive method like near infrared spectroscopy can be advantages and interesting, I suggest authors to make revisions to improve the manuscript.
- It would be helpful to elaborate further on the advantages of using NIR spectroscopy technique. Perhaps add more references that supports the statements regarding its speed and accuracy. Also emphasize why NIR spectroscopy is appropriate for this study rather than other non-destructive methods that can be used.
The following is input in our discussion in red color in revised manuscript
4.6 The merit of this study
We have tried in this work with different modeling methods, featured wavelength selection and using the balanced data but the model performance was not improved. The machine learning algorithms in this study were from 2 linear classifier algorithms (LDA, kNN) and 2 hybrid classifier algorithms (linear and nonlinear ), including SVM and ANN. We concluded that the inherent characteristics of the data set in which the weak correlation between spectral characteristics and the TSI of the medical glove samples might be. Before we end everything, lets recall the chemistry of the bio-polymers, in our case the natural rubber, and its relation to NIR radiation.
Natural rubber is a naturally occurring nanocomposite with an island-nanomatrix structure which is composed of cis-1,4-polyisoprene particles with an average diameter of ~1 μm dispersed in a nanomatrix (several tens of nanometers thick) of nonrubber components such as proteins and phospholipids and the island-nanomatrix structure is stabilized by physical and chemical pinning with proteins and phospholipids that is based on the fact that cis-1,4-polyisoprene of natural rubber is a branched polymer [37].Medical glove is vulcanized natural rubber product where crosslinks must be formed between the polymer chains to provide adequate mechanical resistance for natural rubber latex products [38]. It was proved the cis-1,4-polyisoprene absorbed NIR radiation at 750, 907 and 920 nm in the NIR shortwavelength by Sirisomboon et al [39] and at 1202, 1390, 1420, 1719, 1780, 1884, 2032 and 2218 nm in the NIR longwavelenth by Sirisomboon et al [5] and these absorption correlate well with the chemical constituent in rubber latex, for example, dry rubber content [5,39,40] and total solids content [39,40] and ammonia [41] and with physical parameter such as viscosity [42].
The studies by Lim and Sirisomboon of crosslink density of natural rubber film developed from prevulcanized latex model which was created by PLSR using the spectra scanned by FT-NIR spectrometer [3], the natural rubber thin film model provided the r2, root mean square error of cross validation and bias of 0.65, 4.01%TSI and −0.028%TSI, respectively, using the wavenumber range of 6102–5446.3 cm-1 and 4428–4242.9 cm-1 (1639-1836 nm (included natural rubber absorption bands) and 2258-2357 nm), whereas for the natural rubber thick film model the r2, root mean square error of cross validation and bias were 0.70, 4.00%TSI and −0.006%TSI, respectively, using the wavenumber range of 6102–4597.7 cm-1 (1639-2175 nm (included natural rubber absorption bands)). By the low cost VIS/NIR diode array spectrometer in the wavelength range of 450-1000 nm using fiber optic probe scanned on the thin and thick film, the models for crosslink density indicated by prevulcanisate relaxed modulus (PRM) had poor results indicated by the R2 of calibration and RMSEC of 0.02 and 17.31 × 104 N/m2 and 0.05 and 16.63 × 104 N/m2, respectively [43].
It is proved in our experiment in this report that the longer wavelength range between 960-1650 nm including 3 bands of natural rubber absorption by low cost linear variable filter ultra compact spectrometer (MicroNIR Pro, 1700ES, Viavi, USA) had improved the NIR spectroscopy model by using LDA and kNN algorithms for classification with accuracy of 76% for validation set and 100% for unknown set, respectively. Amirruddin et al. [44] categorized balanced accuracies as poor below 40.00%, moderate within 40.00%–80.00%, and excellent above 80.00%. With 76% accuracy, it is obvious that the best model, constructed before SMOTE using LDA with 2nd derivative spectra and 15 k-Best selected features was moderate model performance and the kNN combined with Savitzky-Golay smoothing + L2เ norm scaling pretreated spectra after SMOTE was excellent with 100% accuracy after tested for 16 unknown samples which was from only G1 (< 80%TSI). This kNN model must be proved with more sample number and more sample groups. Therefore, both developed models can be implemented in the glove factory for screening purpose in the production line.
This might be concluded that this wavelength range spectra (960-1650), i.e. 15 wavelengths by k-Best algorithm and 63 wavelengths by GA, contained crosslink density featured information of the natural rubber thick film of medical gloves.
In term of speed of crosslink density measurement, TSI reference laboratory test take 15 min to get the result and the sample is destroyed while by the non-destructive NIR spectroscopy only 30 s is needed. The NIR spectroscopy is suitable for homogeneous material where NIR hyperspectral image is not necessary. The medical glove is a fairly homogeneous material, therefore, no need to use hyperspectral image technique which the tremendously higher price.
[3] Lim, C. H., & Sirisomboon, P. (2018). Near infrared spectroscopy as an alternative method for rapid evaluation of toluene swell of natural rubber latex and its products. Journal of Near Infrared Spectroscopy, 26(3), 159-168.
[37] Kawahara, S. (2023). Discovery of island-nanomatrix structure in natural rubber. Polymer Journal, 1-15.
[38] de Lima, D. R., Vieira, I. R. S., da Rocha, E. B. D., de Sousa, A. M. F., da Costa, A. C. A., & Furtado, C. R. G. (2023). Biodegradation of natural rubber latex films by highlighting the crosslinked bond. Industrial Crops and Products, 204, 117290.
[39] Sirisomboon, P., Deeprommit, M., Suchaiboonsiri, W., & Lertsri, W. (2013). Shortwave near infrared spectroscopy for determination of dry rubber content and total solids content of Para rubber (Hevea brasiliensis) latex. Journal of Near Infrared Spectroscopy, 21(4), 269-279.
[40] Maraphum, K., Wanjantuk, P., Hanpinitsak, P., Paisarnsrisomsuk, S., Lim, C. H., & Posom, J. (2022). Fast determination of total solids content (TSC) and dry rubber content (DRC) of para rubber latex using near-infrared spectroscopy. Industrial Crops and Products, 187, 115507
[41] Narongwongwattana, S., Rittiron, R., & Hock, L. C. (2015). Rapid determination of alkalinity (ammonia content) in Para rubber latex using portable and Fourier transform-near infrared spectrometers. Journal of Near Infrared Spectroscopy, 23(3), 181-188.
[42] Sirisomboon, P., Chowbankrang, R., & Williams, P. (2012). Evaluation of apparent viscosity of Para rubber latex by diffuse reflection near-infrared spectroscopy. Applied Spectroscopy, 66(5), 595-599.
[43] Lim, C. H., & Sirisomboon, P. (2021). Measurement of cross link densities of prevulcanized natural rubber latex and latex products using low-cost near infrared spectrometer. Industrial Crops and Products, 159, 113016.
[44] Amirruddin, A.D., et al., Hyperspectral remote sensing for assessment of chlorophyll sufficiency levels in mature oil palm (Elaeis guineensis) based on frond numbers: Analysis of decision tree and random forest. Computers and Electronics in Agriculture, 2020. 169: p. 105221.
- The manuscript lacks detailed description of the machine learning algorithms used in this study. Provide brief description of the algorithms/methods with relevant references.
The following is input in the methodology with red color
In order to identify an appropriate model for classifying the crosslink density level of para rubber medical gloves, the supervised machine learning classification algorithms, including Artificial Neural Networks (ANN), Support Vector Machines (SVM), k-Nearest Neighbors (kNN), and Linear Discriminant Analysis (LDA), considering the distinct strengths exhibited by each algorithm type.
Artificial Neural Networks (ANN) are a form of deep learning that models the neural structure of the human brain [20]. They consist of interconnected nodes (neurons) organized into layers (input, hidden, output) [21]. ANN learns by adjusting weights in response to input data, aiming to map inputs to outputs through a training process, commonly employing algorithms such as backpropagation [21]. ANN demonstrates the ability to comprehend complex patterns and adaptability [22].
Support Vector Machines (SVM) are a supervised learning algorithm. They seek a hyperplane that effectively segregates data into distinct classes, aiming to maximize the margin (distance) between the closest points of different classes, referred to as support vectors, for accurate classification of new data points [23]. SVM proves effective in high-dimensional spaces [24].
k Nearest Neighbors (kNN) is a straightforward, instance-based learning algorithm utilized for classification. It predicts outcomes based on the majority class or average value of the k nearest data points to a query point in the feature space [25]. Being non-parametric, it requires no training [26].
Linear Discriminant Analysis (LDA) is a technique used for both dimensionality reduction and classification [27]. It aims to determine linear combinations of features that effectively differentiate classes within a dataset [28]. LDA projects data onto a lower-dimensional space, maximizing the distance between class means and minimizing the variance within each class [28]. It functions as a dimensionality reduction method and effectively handles multi-class problems.
[20] Walczak, S. and N. Cerpa, Artificial Neural Networks, in Encyclopedia of Physical Science and Technology 2003, Elsevier Science Ltd. p. 631-645.
[21] Mohseni-Dargah, M., et al., Machine learning in surface plasmon resonance for environmental monitoring, in Artificial Intelligence and Data Science in Environmental Sensing. 2022, Elsevier Inc. p. 269-298.
[22] Sadiq, R., M.J. Rodriguez, and H.R. Mian, Empirical Models to Predict Disinfection By-Products (DBPs) in Drinking Water: An Updated Review, in Encyclopedia of Environmental Health (Second Edition). 2019, Elsevier B.V. p. 324-338.
[23] Whittingham, H. and S.K. Ashenden, Hit discovery, in The Era of Artificial Intelligence, Machine Learning, and Data Science in the Pharmaceutical Industry. 2021, Elsevier Inc. p. 81-102.
[24] Gove, R. and J. Faytong, Machine Learning and Event-Based Software Testing: Classifiers for Identifying Infeasible GUI Event Sequences, in Advances in Computers. 2012, Elsevier Inc. p. 109-135.
[25] Jeffers, J., J. Reinders, and A. Sodani, Machine learning, in Intel Xeon Phi Processor High Performance Programming. 2016, Elsevier Inc. p. 527-548.
[26] Shi, Y., et al., Primer on artificial intelligence, in Mobile Edge Artificial Intelligence. 2022, Elsevier Inc. p. 7-36.
[27] Mohanty, N., et al., Shape-Based Image Classification and Retrieval, in Handbook of Statistics. 2013, Elsevier B.V. . p. 249-267.
[28] Vaibhaw, J. Sarraf, and P.K. Pattnaik, Brain–computer interfaces and their applications, in An Industrial IoT Approach for Pharmaceutical Industry Growth. 2020, Elsevier Inc. p. 31-54.
- I suggest using different line types for different TSI in Figure 3 for better legibility.
It isn’t good to change the line types. So, we keep it in the present format with reduced line-width.
(a)
(b)
(c)
(d)
(e)
(f)
(g)
(h)
(i)
(j)
Figure 3. The (a) raw; (b) Savitzky - Golay smoothing; (c) 2nd derivative; (d) multiplicative scatter correction (MSC); (e) standard normal variate (SNV); (f) detrending; (g) min-max scaling; (h) robust normal variate (RNV); (i) log transform; (j) L2 norm scaling spectra of different levels of toluene swell index (Blue TSI less than 80 %, Black TSI 80-88 %, and Red TSI more than 88 %).
- Provide further explanation on why LDA achieved the highest accuracy during calibration however exhibited the lowest accuracy after validation.
The following is input in the Results part with red color
3.3.1 Full spectra
The results of classifying three different TSI levels of thick-film medical gloves for the calibration and prediction sets in an imbalanced dataset were presented in Table 4 which shows the LDA achieved the highest accuracy (0.99) during calibration, but when validated, it exhibited the lowest accuracy (0.46). In contrast, other models demonstrated similar performance for both the calibration and validation sets. The calibration accuracies of ANN, SVM, and kNN were 0.70, 0.74, and 0.77, respectively, while the validation set accuracies were 0.84, 0.70, and 0.70, respectively. For LDA, though the accuracy in calibration was the highest but the accuracy in validation was the lowest indicated the overfit of the model which occurred due to the sample size was not significantly conformed to the the number of hyperparameters tuned [30,31] and to the effect of the separation method which caused the distribution of the pretreated spectra of the calibration set and validation set different. This can be used for rationale for the LDA models in Table 5-7.
[30] Ludwig, B., Murugan, R., Parama, V. R., & Vohland, M. (2019). Accuracy of estimating soil properties with mid‐infrared spectroscopy: Implications of different chemometric approaches and software packages related to calibration sample size. Soil Science Society of America Journal, 83(5), 1542-1552.
[31] Cawley, G.C., and Talbot, N.L.C.. 2010. On over-fitting in model selection and subsequent selection bias in performance evaluation. J. Mach. Learn. Res. 11: 2079–2107.
- Provide relevant references for the data processing methods used in this study, for example Savitzky - Golay smoothing.
The following is input in the methodology part in red color
The pretreated methods of Savitzky - Golay smoothing resulted in reduced noise of the spectrum was first applied and then the second derivative (segment size 21) [9], multiplicative scatter correction (MSC), standard normal variate (SNV), detrending, and normalization were applied consecutively [10].
Other preprocessing from normalization, including min-max normalization is also considered in this study. Besides that, robust normal variate (RNV) preprocessing is also used in this study to handle light scatter effects like SNV. If the SNV formula is subtraction by the mean, RNV is subtraction by the median of each spectral variable and subsequently dividing that value by the standard deviation of the spectrum. Like SNV, RNV and L2 norm scaling can also handle scatter problems on the spectrum [11]. Finally, log transformation preprocessing is also used to scale and transform NIR spectra to increase the relationship between absorbance and response to become linear again [12].
[9] Yu, D.-x., Guo, S., Zhang, X., Yan, H., Zhang, Z.-y., Chen, X., Chen, J.-y., Jin, S.-j., Yang, J., & Duan, J.-a. (2022). Rapid detection of adulteration in powder of ginger (Zingiber officinale Roscoe) by FT-NIR spectroscopy combined with chemometrics. Food Chemistry: X, 15, 100450.
[10] Torniainen, J., Afara, I. O., Prakash, M., Sarin, J. K., Stenroth, L., & Töyräs, J. (2020). Open-source python module for automated preprocessing of near infrared spectroscopic data. Analytica Chimica Acta, 1108, 1-9.
[11] Engel, J., Gerretzen, J., Szymańska, E., Jansen, J. J., Downey, G., Blanchet, L., & Buydens, L. M. C. (2013). Breaking with trends in pre-processing? TrAC Trends in Analytical Chemistry, 50, 96-106.
[12] Mallet, A., Tsenkova, R., Muncan, J., Charnier, C., Latrille, É., Bendoula, R., Steyer, J.-P., & Roger, J.-M. (2021). Relating Near-Infrared Light Path-Length Modifications to the Water Content of Scattering Media in Near-Infrared Spectroscopy: Toward a New Bouguer–Beer–Lambert Law. Analytical Chemistry, 93(17), 6817-6823.
Comments on the Quality of English Language
The overall quality of English in this manuscript is very poor; current manuscript contains a lot of grammatical errors and typos, which makes it difficult to follow. The authors must carefully review and revise the entire manuscript for better English.
We will send our manuscript if accepted by Polymers to MDPI English Service

Reviewer 4 Report
Comments and Suggestions for Authors
In this work, the authors try to classify the crosslink density level of para rubber film of medical glove by using near-infrared spectral data. The reviewer believes that a combination of machine learning to experiments are not bad, however, it is confusing that whether all the machine learnings are really necessary and what it brings to us. Here the size of samples are not large, and the learning process is also one of the common routines (in principle it can be applied to anything), without deep physical meanings and modeling behind it. Therefore, the reviewer believe this work is not suitable for Polymers.
Some issues maybe helpful:
Figure 1 does not show any useful information and should be removed.
The size of samples are not large. It is confusing that machine learning is really useful or not.
More theory and discussion from the polymer chemistry aspect should be applied. Otherwise it is more or less out of the topic.
It is benefit to measure the density with experimental method and compare it to the model-predicted results.
Comments on the Quality of English Language
Moderate editing of English language required
Author Response
Reviewer 4
In this work, the authors try to classify the crosslink density level of para rubber film of medical glove by using near-infrared spectral data. The reviewer believes that a combination of machine learning to experiments are not bad, however, it is confusing that whether all the machine learnings are really necessary and what it brings to us. Here the size of samples are not large, and the learning process is also one of the common routines (in principle it can be applied to anything), without deep physical meanings and modeling behind it. Therefore, the reviewer believe this work is not suitable for Polymers.
Some issues maybe helpful:
Figure 1 does not show any useful information and should be removed.
It is removed
The size of samples are not large. It is confusing that machine learning is really useful or not.
We would like to explain that we have not used deep learning like CNN which a huge number of data is necessary. Also, in this study, we have considered 2 algorithms as linear classifiers (LDA, KNN) and 2 algorithms as hybrid classifiers (linear and nonlinear ), including SVM and ANN, with the total real number of spectra in this study is 130, of which 93 (42:29:22) were used as training and 37 as testing. With the SMOTE method, the total spectra were increased to 163, of which 126 (42:42:42) were used as training, and 37 real spectra were used as testing. The samples collection duration of our experiment was during 28 January 2022 till 9 January 2023 which is the year-round production of the factory, though less in your opinion but this confirm the robustness with wide variation data of our calibration model.
However, it's important to note that sample collection is a time-consuming process. Within this study, we managed to collect 130 samples over the span of one year. Gathering additional samples presents a difficulty due to the significant time investment it demands.
The following is input in discussion in the small sample number issue in red color
4.4 Effect of sample number
By the principle that the model performance will be better , if the the number of samples is large due to the small error. In case of ANN, the number of data values used for training must exceed that of weights determined in the network; this entails using a large number of samples for calibration if the number of input variables is also large. Based on the results, PC-ANN, where the data dimension was reduced, was the best choice for the intended application [33], in which our case we used GA-ANN, k Best and PC-ANN where the variables was reduced from the original of full spectra to some featured wavelength data. Recently in 2023, Rasooli Sharabiani et al [34] used ANN with samples of winter wheat leaf for evaluation of chlorophyll content based on VIS/NIR spectroscopy using PLSR and ANN where 120 samples was for training set and the left was for test set and the models result the most accurate prediction, with the correlation coefficient of 0.92 and 0.97, along with the root mean square error of 0.9131 and 0.7305 respectively. Ni et al [35] suggested that back propagation ANN (BANN) were powerful and promising methods for handling linear as well as nonlinear systems, even when the data sets are moderately small and they indicated that when very little data is available BANN had an additional advantage of achieving robust predictive performance based on relatively small data sets compared to other nonlinear approaches while being less influenced by preprocessing, i.e. SNV.
[33] Blanco, M., Coello, J., Iturriaga, H., Maspoch, S., & Pages, J. (2000). NIR calibration in non-linear systems: different PLS approaches and artificial neural networks. Chemometrics and Intelligent Laboratory Systems, 50(1), 75-82.
[34] Rasooli Sharabiani, V., Soltani Nazarloo, A., Taghinezhad, E., Veza, I., Szumny, A., & Figiel, A. (2023). Prediction of winter wheat leaf chlorophyll content based on VIS/NIR spectroscopy using ANN and PLSR. Food Science & Nutrition, 11(5), 2166-2175.
[35] Ni, W., Nørgaard, L., & Mørup, M. (2014). Non-linear calibration models for near infrared spectroscopy. Analytica chimica acta, 813, 1-14.
More theory and discussion from the polymer chemistry aspect should be applied. Otherwise it is more or less out of the topic.
The polymer chemistry of our samples was discussed and input in the discussion part in red color as follows:
4.6 The merit of this study
We have tried in this work with different modeling methods, featured wavelength selection and using the balanced data but the model performance was not improved. The machine learning algorithms in this study were from 2 linear classifier algorithms (LDA, kNN) and 2 hybrid classifier algorithms (linear and nonlinear ), including SVM and ANN. We concluded that the inherent characteristics of the data set in which the weak correlation between spectral characteristics and the TSI of the medical glove samples might be. Before we end everything, lets recall the chemistry of the bio-polymers, in our case the natural rubber, and its relation to NIR radiation.
Natural rubber is a naturally occurring nanocomposite with an island-nanomatrix structure which is composed of cis-1,4-polyisoprene particles with an average diameter of ~1 μm dispersed in a nanomatrix (several tens of nanometers thick) of nonrubber components such as proteins and phospholipids and the island-nanomatrix structure is stabilized by physical and chemical pinning with proteins and phospholipids that is based on the fact that cis-1,4-polyisoprene of natural rubber is a branched polymer [37].Medical glove is vulcanized natural rubber product where crosslinks must be formed between the polymer chains to provide adequate mechanical resistance for natural rubber latex products [38]. It was proved the cis-1,4-polyisoprene absorbed NIR radiation at 750, 907 and 920 nm in the NIR shortwavelength by Sirisomboon et al [39] and at 1202, 1390, 1420, 1719, 1780, 1884, 2032 and 2218 nm in the NIR longwavelenth by Sirisomboon et al [5] and these absorption correlate well with the chemical constituent in rubber latex, for example, dry rubber content [5,39,40] and total solids content [39,40] and ammonia [41] and with physical parameter such as viscosity [42].
The studies by Lim and Sirisomboon of crosslink density of natural rubber film developed from prevulcanized latex model which was created by PLSR using the spectra scanned by FT-NIR spectrometer [3], the natural rubber thin film model provided the r2, root mean square error of cross validation and bias of 0.65, 4.01%TSI and −0.028%TSI, respectively, using the wavenumber range of 6102–5446.3 cm-1 and 4428–4242.9 cm-1 (1639-1836 nm (included natural rubber absorption bands) and 2258-2357 nm), whereas for the natural rubber thick film model the r2, root mean square error of cross validation and bias were 0.70, 4.00%TSI and −0.006%TSI, respectively, using the wavenumber range of 6102–4597.7 cm-1 (1639-2175 nm (included natural rubber absorption bands)). By the low cost VIS/NIR diode array spectrometer in the wavelength range of 450-1000 nm using fiber optic probe scanned on the thin and thick film, the models for crosslink density indicated by prevulcanisate relaxed modulus (PRM) had poor results indicated by the R2 of calibration and RMSEC of 0.02 and 17.31 × 104 N/m2 and 0.05 and 16.63 × 104 N/m2, respectively [43].
It is proved in our experiment in this report that the longer wavelength range between 960-1650 nm including 3 bands of natural rubber absorption by low cost linear variable filter ultra compact spectrometer (MicroNIR Pro, 1700ES, Viavi, USA) had improved the NIR spectroscopy model by using LDA and kNN algorithms for classification with accuracy of 76% for validation set and 100% for unknown set, respectively. Amirruddin et al. [44] categorized balanced accuracies as poor below 40.00%, moderate within 40.00%–80.00%, and excellent above 80.00%. With 76% accuracy, it is obvious that the best model, constructed before SMOTE using LDA with 2nd derivative spectra and 15 k-Best selected features was moderate model performance and the kNN combined with Savitzky-Golay smoothing + L2เ norm scaling pretreated spectra after SMOTE was excellent with 100% accuracy after tested for 16 unknown samples which was from only G1 (< 80%TSI). This kNN model must be proved with more sample number and more sample groups. Therefore, both developed models can be implemented in the glove factory for screening purpose in the production line.
This might be concluded that this wavelength range spectra (960-1650), i.e. 15 wavelengths by k-Best algorithm and 63 wavelengths by GA, contained crosslink density featured information of the natural rubber thick film of medical gloves.
In term of speed of crosslink density measurement, TSI reference laboratory test take 15 min to get the result and the sample is destroyed while by the non-destructive NIR spectroscopy only 30 s is needed. The NIR spectroscopy is suitable for homogeneous material where NIR hyperspectral image is not necessary. The medical glove is a fairly homogeneous material, therefore, no need to use hyperspectral image technique which the tremendously higher price.
[3] Lim, C. H., & Sirisomboon, P. (2018). Near infrared spectroscopy as an alternative method for rapid evaluation of toluene swell of natural rubber latex and its products. Journal of Near Infrared Spectroscopy, 26(3), 159-168.
[37] Kawahara, S. (2023). Discovery of island-nanomatrix structure in natural rubber. Polymer Journal, 1-15.
[38] de Lima, D. R., Vieira, I. R. S., da Rocha, E. B. D., de Sousa, A. M. F., da Costa, A. C. A., & Furtado, C. R. G. (2023). Biodegradation of natural rubber latex films by highlighting the crosslinked bond. Industrial Crops and Products, 204, 117290.
[39] Sirisomboon, P., Deeprommit, M., Suchaiboonsiri, W., & Lertsri, W. (2013). Shortwave near infrared spectroscopy for determination of dry rubber content and total solids content of Para rubber (Hevea brasiliensis) latex. Journal of Near Infrared Spectroscopy, 21(4), 269-279.
[40] Maraphum, K., Wanjantuk, P., Hanpinitsak, P., Paisarnsrisomsuk, S., Lim, C. H., & Posom, J. (2022). Fast determination of total solids content (TSC) and dry rubber content (DRC) of para rubber latex using near-infrared spectroscopy. Industrial Crops and Products, 187, 115507
[41] Narongwongwattana, S., Rittiron, R., & Hock, L. C. (2015). Rapid determination of alkalinity (ammonia content) in Para rubber latex using portable and Fourier transform-near infrared spectrometers. Journal of Near Infrared Spectroscopy, 23(3), 181-188.
[42] Sirisomboon, P., Chowbankrang, R., & Williams, P. (2012). Evaluation of apparent viscosity of Para rubber latex by diffuse reflection near-infrared spectroscopy. Applied Spectroscopy, 66(5), 595-599.
[43] Lim, C. H., & Sirisomboon, P. (2021). Measurement of cross link densities of prevulcanized natural rubber latex and latex products using low-cost near infrared spectrometer. Industrial Crops and Products, 159, 113016.
[44] Amirruddin, A.D., et al., Hyperspectral remote sensing for assessment of chlorophyll sufficiency levels in mature oil palm (Elaeis guineensis) based on frond numbers: Analysis of decision tree and random forest. Computers and Electronics in Agriculture, 2020. 169: p. 105221.
It is benefit to measure the density with experimental method and compare it to the model-predicted results.
We then did experiment using 16 real product of medical gloves of 4 factories
We then did experiment using 16 real product of medical gloves of 4 factories
2.6 Validation by unknown real sample set form factories
Medical rubber gloves without powder was collected from 4 factories (4 gloves for 1 factory). Therefore, 16 gloves. Each glove was scanned by placing MicroNIR window on the intact glove (two-layer scan) placed on the aluminum plate as a reflector make this scanning the transflectance mode. The scanning was done 5 scans per position and 4 positions on one glove. Therefore, 320 spectra in total. Then, the MicroNIR was inserted inside the glove and scanned the glove only one layer by the same procedure. After that the glove was subjected to the TSI test immediately.
Every spectrum scanned was subjected to some models developed including LDA, kNN and SVM with different pretreatment methods and the model classification performance were calculated.
3.1.4 Validation result by unknown real sample set form factories
Table 10 shows the production information of the unknown glove samples and the TSI value of every sample. The models developed were used to predict the TSI value of the glove sample. The TSI of every glove was 60% except one was 72% which was in group 1 (<80%). This indicated the uniformity of the gloves production. Though, there may have an opportunity to have out of accetable group indicating the need of non-destructive detection of the product real time on-line for 100% crossling density levels detection.
By TSI test, the result show that every glove product was in group 1 where TSI was less than 80%. Table 11 show the prediction results of the high performance models from modelling state. Unexpectedly. The best model by LDA with second derivative spectral pretreatment + k-best wavelength selection could not predict accurately but KNN developed by full spectra and Savitzky-Golay smoothing + L2 norm scaling pretreatment + GA wavelength selection spectra provided 100% accuracy for both scan on one-layer or two-layer scan but using the data after SMOTE (Table 11). It was observed that the results of most of the models show similar accuracy when scan by both scan-layer incating no prediction problem of one-layer model by scanning of double layer glove.
Figure 6 shows the raw spectra (Figure 6 (a)) and the pretreated spectra by Savitzky-Golay smoothing + L2 norm scaling (Figure 6 (b) of the gloves in unknown samples both one-layer and two-layer scan. The raw spectra shows the baseline shift and tillting effects due to physical factors such as density while the same peaks illustraed the same constituents. After pretreated, the baseline effect was mostly eliminated and the hight of peaks could inform the different radiation absorption due to different quantity of constituents.
Table 10. The information of unknown medical glove product from different factories
|
Factory |
production date |
expired date |
initial diameter (mm) |
% TSI |
|
sl1 |
08-2022 |
08-2025 |
40 |
60 |
|
sl2 |
08-2022 |
08-2025 |
40 |
60 |
|
sl3 |
08-2022 |
08-2025 |
40 |
60 |
|
sl4 |
08-2022 |
08-2025 |
40 |
60 |
|
PO1 |
05-2023 |
05-2026 |
40 |
60 |
|
PO2 |
05-2023 |
05-2026 |
40 |
60 |
|
PO3 |
05-2023 |
05-2026 |
40 |
60 |
|
PO4 |
05-2023 |
05-2026 |
40 |
60 |
|
PX 1 |
10-2023 |
10-2026 |
40 |
60 |
|
PX 2 |
10-2023 |
10-2026 |
40 |
60 |
|
PX 3 |
10-2023 |
10-2026 |
40 |
60 |
|
PX 4 |
10-2023 |
10-2026 |
40 |
60 |
|
St1 |
09-2023 |
09-2026 |
43 |
72 |
|
St2 |
09-2023 |
09-2026 |
40 |
60 |
|
St3 |
09-2023 |
09-2026 |
40 |
60 |
|
St4 |
09-2023 |
09-2026 |
40 |
60 |
Table 11. Accuracy of classification of toluene swell index level of unknown samples by developled models using the before SMOTE and after SMOTE.
|
Full Spectrum |
Best preprocessing + algorithm |
Scanning method |
Group |
||
|
G1 (TSI < 80) |
G2 (80<TSI<88) |
G3 (TSI > 88) |
|||
|
Before SMOTE |
(Savitzky-Golay smoothing + RNV) + SVM |
One-layer scan |
288 |
12 |
20 |
|
Two-layer scan |
276 |
21 |
23 |
||
|
After SMOTE |
(Savitzky-Golay smoothing + L2 norm scaling) + kNN |
One-layer scan |
320 |
0 |
0 |
|
Two-layer scan |
320 |
0 |
0 |
||
|
Selection wavelength |
Best preprocessing + algorithm |
Scanning method |
Group |
||
|
1 (TSI < 80) |
2 (80<TSI<88) |
3 (TSI > 88) |
|||
|
Before SMOTE |
(Second derivative) + k-best + LDA |
One-layer scan |
55 |
1 |
264 |
|
Two-layer scan |
263 |
45 |
12 |
||
|
After SMOTE |
(Savitzky-Golay smoothing + L2เ norm scaling) + GA + kNN |
One-layer scan |
320 |
0 |
0 |
|
Two-layer scan |
320 |
0 |
0 |
||
|
Reduction features |
Best preprocessing + algorithm |
Scanning method |
Group |
||
|
1 (TSI < 80) |
2 (80<TSI<88) |
3 (TSI > 88) |
|||
|
Before SMOTE |
(Savitzky-Golay smoothing + Detrending) + 10-PC + LDA |
One-layer scan |
128 |
104 |
88 |
|
Two-layer scan |
137 |
104 |
79 |
||
|
After SMOTE |
(Savitzky-Golay smoothing + RNV) + 10-PC + LDA |
One-layer scan |
141 |
69 |
110 |
|
Two-layer scan |
127 |
92 |
101 |
||
(a)
(b)
Figure 6 The (a) raw spectra and (b) the pretreated spectra by Savitzky-Golay smoothing + L2 norm scaling of the gloves in unknown samples both one-layer and two-layer scan.
Comments on the Quality of English Language
Moderate editing of English language
We will send our manuscript if accepted by Polymers to MDPI English Service

Round 2
Reviewer 4 Report
Comments and Suggestions for Authors
Since the authors provided much more information and explanations, as well as experimental results, I believe this work meats the standard for publication. Thank you for your efforts.
Author Response
Dear Editor
Thanks so much for your kind consideration to let us revise our manuscript.
Decision
Accept after minor revision
Comments
Dear Authors, I recommend your corrected version of manuscript for publication, but after minor revision.
Thanks so much.
For me, the main problem it is the lack of comments to Figures, particularly to Figs 4, 5, 6. There is only mentioned (line 533) about Figs 4 and 5,while Figure 6 is without any comment.
Please add comments to results presented in Figures. Only the caption under figure, without any discussion of presented results is really not correct.
In your manuscript only Figure 3 has enough comment (line 446).
We have revised followed Editor comment
We have used Yellow mark and Black - letters for our revision
Figure 4 (Line 403 - 408) in Result part of the revised manuscript
From Figure 4, PC1 have covered the highest spectral informative variance of 94.42% left only 4.77 and 0.63% for PC2 and PC3 and 0.18% for the PC4 to PC125. However, when PC score of diiferent PCs were used for model development where the PC1 was the main PC could only classify with the performance of 0.75-0.76 indicating though the highest spectral informative variance but there were just moderate correlation between those spectra information.
Figure 5 (Line 409-415) in Result part of the revised manuscript
These performance indicators correlation together with the X-loading loading of PC1 to PC3 (Figure 5) indicated the moderate relationship between the PC score of NIR spectral data obtained by PCA and TSI. From Figure 5, the high peaks of cis-1,4-polyisoprene from pure Para rubber sheet at 1200, 1390 and 1420 nm as indicated by Sirisomboon, et al. [5] were shown in the X-loading loading of PC1 to PC3 confirmed that there were the relationship between the NIR vibration of natural rubber with the related property in which this case was TSI but moderate.
Figure 6 (Line 439 – 447) in Result part of the revised manuscript
Figure 6 shows the raw spectra (Figure 6 (a)) and the pretreated spectra by Savitzky-Golay smoothing + L2 norm scaling (Figure 6 (b) of the gloves in unknown samples both one-layer and two-layer scan. The raw spectra shows the same peaks as the raw spectra of the modeling set but the peaks between 1400-1500 nm were shifted slightly to the left due to slightly different in production process of the factories from ours. There were the baseline shift and tillting effects due to physical factors such as density while the same peaks illustrated the same constituents. After pretreated, the baseline effect was mostly eliminated and the hight of peaks could inform the different radiation absorption due to different quantity of constituents.
We have input information essential for MicroNIR scanning in revised manuscript as following
2.2 NIR Spectroscopy
Ultra compact NIR spectrometer with wavelength range of 900 -1700 nm (MicroNIR Pro 1700ES Spectrometer, VIAVI Solutions Inc., California, USA) was used for absorbance spectrum of the thick film sample acquisition. The scanning resolution was 6.2 nm. Therefore, there were 125 points (908-1676 nm) obtained to form spectrum. The white reference spectrum and dark reference spectrum for the background compensation were scanned in the beginning of every 10 samples to be scanned. The white reference material was Spectralon® and the dark reference spectrum was obtained by scanning the flour from the height of ~ 60 cm.
Please kindly determine for the possibility of our revised manuscript can be published.
Respectfully Yours.
Jetsada Posom
